# Hydrop enables droplet-based single-cell ATAC-seq and single-cell RNA-seq using dissolvable hydrogel beads

**Florian V De Rop**[1,2], **Joy N Ismail**[1,2], **Carmen Bravo González-Blas**[1,2], **Gert J Hulselmans**[1,2], **Christopher Campbell Flerin**[1,2,3], **Jasper Janssens**[1,2], **Koen Theunis**[1,2,3], **Valerie M Christiaens**[1,2], **Jasper Wouters**[1,2], **Gabriele Marcassa**[1,4], **Joris de Wit**[1,4], **Suresh Poovathingal**[1]\*†, **Stein Aerts**[1,2]\*†

[1]VIB-KU Leuven/VIB Center for Brain & Disease Research, Leuven, Belgium; [2]Laboratory of Computational Biology, Department of Human Genetics, KU Leuven, Leuven, Belgium; [3]Aligning Science Across Parkinson's (ASAP) Collaborative Research Network, Chevy Chase, United States; [4]Laboratory of Synapse Biology, Department of Neurosciences, KU Leuven, Leuven, Belgium

**\*For correspondence:**
suresh.poovathingal@kuleuven.be (SP);
stein.aerts@kuleuven.be (SA)

†These authors contributed equally to this work

**Abstract** Single-cell RNA-seq and single-cell assay for transposase-accessible chromatin (ATAC-seq) technologies are used extensively to create cell type atlases for a wide range of organisms, tissues, and disease processes. To increase the scale of these atlases, lower the cost and pave the way for more specialized multiome assays, custom droplet microfluidics may provide solutions complementary to commercial setups. We developed HyDrop, a flexible and open-source droplet microfluidic platform encompassing three protocols. The first protocol involves creating dissolvable hydrogel beads with custom oligos that can be released in the droplets. In the second protocol, we demonstrate the use of these beads for HyDrop-ATAC, a low-cost noncommercial scATAC-seq protocol in droplets. After validating HyDrop-ATAC, we applied it to flash-frozen mouse cortex and generated 7996 high-quality single-cell chromatin accessibility profiles in a single run. In the third protocol, we adapt both the reaction chemistry and the capture sequence of the barcoded hydrogel bead to capture mRNA, and demonstrate a significant improvement in throughput and sensitivity compared to previous open-source droplet-based scRNA-seq assays (Drop-seq and inDrop). Similarly, we applied HyDrop-RNA to flash-frozen mouse cortex and generated 9508 single-cell transcriptomes closely matching reference single-cell gene expression data. Finally, we leveraged HyDrop-RNA's high capture rate to analyze a small population of fluorescence-activated cell sorted neurons from the *Drosophila* brain, confirming the protocol's applicability to low input samples and small cells. HyDrop is currently capable of generating single-cell data in high throughput and at a reduced cost compared to commercial methods, and we envision that HyDrop can be further developed to be compatible with novel (multi) omics protocols.

## Editor's evaluation

De Rop et al. introduce a flexible microfluidics-based single-cell genomics technology that expands and improves previously existing custom droplet-based scRNA-seq protocols (inDrops and Drop-seq) in interesting directions: better data quality, simplified workflow, high-cell recovery, and flexibility towards other single-cell applications.

**eLife digest** Scientists are now able to determine the order of chemical blocks, or nucleic acids, that make up the genetic code. These sequencing tools can be used to identify which genes are active within a biological sample. They do this by extracting and analysing open chromatin (regions of DNA that are accessible to the cell's machinery), or sequences of RNA (the molecular templates cells use to translate genes into working proteins).

Initially, most sequencing tools could only provide an 'averaged-out' profile of the genes activated in bulk pieces of tissue which contain multiple types of cell. However, advances in technology have led to new methods that can extract and analyse open chromatin or RNA from individual cells.

First, the cells are separated, via a technique called microfluidics, into tiny droplets of water along with a single bead that carries a unique barcode. The cell is then broken apart inside the droplet and the barcode within the bead gets released and attaches itself to the genetic material extracted from the cell. All the genetic material inside the droplets is then pooled together and sequenced. Researchers then use the barcode tags to identify which bits of RNA or DNA belong to each cell.

Single-cell sequencing has many advantages, including being able to pinpoint precise genetic differences between healthy and abnormal cells, and to create cell atlases of whole organisms, tissues and microbial communities. But existing methods for extracting chromatin are very expensive, and there were no openly available tools for processing thousands of cells at speed. Furthermore, while several single-cell RNA sequencing tools are already freely available, they are not very sensitive or practical to use.

Here, De Rop et al. have developed a new open-source platform called HyDrop that overcomes these barriers. The method entails a new type of barcoded bead and optimised elements of existing microfluidics protocols using open-source reagents. These changes created a more user-friendly workflow and increased sensitivity of sequencing at no additional cost.

De Rop et al. used their new platform to screen the RNA and open chromatin of thousands of individuals cells from the brains of mice and flies. HyDrop outperformed other open-source methods when working in RNA-sequencing mode. It also provides the first open-source tool for sequencing open chromatin in single cells. Further improvements are expected as researchers tweak the platform, which for now provides an affordable alternative to existing methods.

## Introduction

In the past 5 years, droplet microfluidics have been applied extensively to partition single cells and sequence their nucleic acid content. Two methods that pioneered the field, inDrop (*Klein et al., 2015*) and Drop-seq (*Macosko et al., 2015*), both rely on the same working principle: individual cells are rapidly encapsulated into a nanoliter droplet together with a barcoded bead. Barcoding primers carried by the beads inside the emulsion are then used to index each individual cell's mRNA. This process occurs either inside the droplet, where the cell's mRNA is reverse transcribed using barcoded primers released by the barcoded bead (inDrop) or after emulsion breaking, where the cellular mRNA is anchored onto barcodes carried by resin beads (Drop-seq). The thousands of single-cell transcriptomes can then be processed for next-generation sequencing in a pooled manner. The high throughput and low per-cell reagent consumption associated with this approach have allowed researchers to profile gene expression of tens of thousands of single cells (*Kalish et al., 2020*; *Yap et al., 2021*; *Karaiskos et al., 2017*) at an affordable cost. However, the labor requirements of current open-source droplet microfluidic protocols combined with their limited sensitivity have hindered widespread adoption. Furthermore, academic development of droplet-microfluidic single-cell sequencing technology after inDrop and Drop-seq has been limited. To our knowledge, only one noncommercial droplet-based single-cell assay for transposase accessible chromatin (scATAC-seq) protocol has been published so far (*Chen et al., 2019*). Despite its elegant conceptual solution to droplet-based combined scATAC and scRNA-seq, the SNARE-seq protocol is still labor intensive, and the use of resin beads leads to reduced cell capture rates. These resin beads are loaded at dilute concentrations to prevent microfluidic obstruction, and as a result, many droplets are empty, leading to a low cell capture rate (~2%), and reagent waste (*Zhang et al., 2019*). Instead, deformable hydrogel beads can be stacked and loaded into droplets at a fixed rate without the risk of microfluidic failure, thereby increasing cell capture rate

to >50% (*Zhang et al., 2019*; *Abate et al., 2009*). Several commercial solutions have emerged since inDrop and Drop-seq (*Zheng et al., 2017*; *Sarkar et al., 2018*), and their application has been used to generate hundreds of thousands (*Davie et al., 2018*; *Svensson et al., 2020*) - and recently millions (*Datlinger et al., 2021*; *Ren et al., 2021*) - of single-cell transcriptomes at a high sensitivity. However, the high cost of these commercial protocols remains prohibitive for many research applications, and their fixed nature limits custom protocol development.

In order to increase the sensitivity and user-friendliness of open-source microfluidic protocols, and to provide a more flexible and open platform, we developed HyDrop, a new hydrogel-based droplet microfluidic method for high-throughput scRNA-seq or scATAC-seq. We adapted inDrop's original isothermal extension bead barcoding protocol (*Zilionis et al., 2017*) to a linear amplification workflow to generate more uniformly barcoded beads. Next, we applied a custom hydrogel bead production process similar to a recently published protocol (*Wang et al., 2020*) to generate dissolvable beads, improving barcoded primer release and diffusion. We also optimized Drop-seq's pooled template-switching reverse transcription strategy for application inside the cell/bead emulsion, and optimized the assay's sensitivity by testing several different cDNA library preparation strategies. The combination of these adaptations resulted in a significantly increased sensitivity for HyDrop-RNA compared to inDrop and Drop-seq at no additional cost, and in a more user-friendly workflow. Additionally, the dissolution of the hydrogel beads stabilizes the cell/bead emulsion during linear amplification thermocycling. This change allowed us to implement to our knowledge the first open-source single-cell ATAC-seq in droplets using hydrogel beads. We applied both technologies to mouse cerebral cortex and generated single-cell ATAC-seq and single-cell RNA-seq data that is highly concordant with reference data.

## Results

## Generation of dissolvable hydrogel beads with barcodes for scATAC-seq and scRNA-seq

We generated barcoded hydrogel beads that can dissolve and release their embedded barcoded oligonucleotide. Polyacrylamide beads incorporating disulfide cross-linkers and short oligonucleotide polymerase chain reaction (PCR) priming sites were generated by droplet microfluidics similar to a recently published method (*Wang et al., 2020*). A custom droplet microfluidic chip (*Figure 3—figure supplement 1*) was employed to produce beads of approximately 50 μm diameter. These hydrogel beads were then barcoded using a modified three-round split-pool linear amplification synthesis method (*Klein et al., 2015*; *Zilionis et al., 2017*), resulting in 96 × 96 × 96 (884,736) barcode possibilities (see *Supplementary file 1* – 'Molecular sequence description of HyDrop bead barcoding' for an in-depth visualization of the nucleic acid sequences in every step). The terminal sequence used in the final round of barcoding can be varied depending on the assay the beads will be used for (see Methods, *Figure 1a*). A sequence complementary to the Tn5 transposase adapter was used to capture tagmented chromatin fragments in scATAC-seq and a unique molecular identifier (UMI) (*Kivioja et al., 2011*)+ poly (dT) sequence was used to capture and count poly (A)+ mRNA in scRNA-seq (protocols described further). The barcoded beads were stored in a glycerol-based freezing buffer at –80°C in order to prevent loss of primers over time (*Figure 1—figure supplement 1*).

We validated the extension of the hydrogel bead primers using fluorescent probes complementary to the beads 3-prime terminal sequence (*Zilionis et al., 2017*; *Figure 1b* and *Figure 1—figure supplement 2*) and the sub-barcode purity using fluorescent probes complementary to 1 of the 96 sub-barcode possibilities (*Figure 1c*, *Figure 1—figure supplements 2 and 3*, Tables S12). These experiments showed that there was no significant loss of primers or mixing of barcodes throughout the barcoding process, and that the beads are uniform in size and primer content. Additional testing revealed that our modified linear amplification barcoding method produced more uniformly barcoded beads compared to the isothermal amplification protocol described in inDrop (*Figure 1—figure supplement 2*). Furthermore, the disulfide moieties incorporated in both the bead's polymer matrix and oligonucleotide linker are cleaved when exposed to reducing conditions, such as dithiothreitol (DTT). This chemical method of release is more user-friendly compared to the UV-mediated (*Klein et al., 2015*; *Zilionis et al., 2017*) primer release as the beads do not have to be shielded from light. In addition to improved primer release compared to nondissolvable beads (*Figure 1—figure*

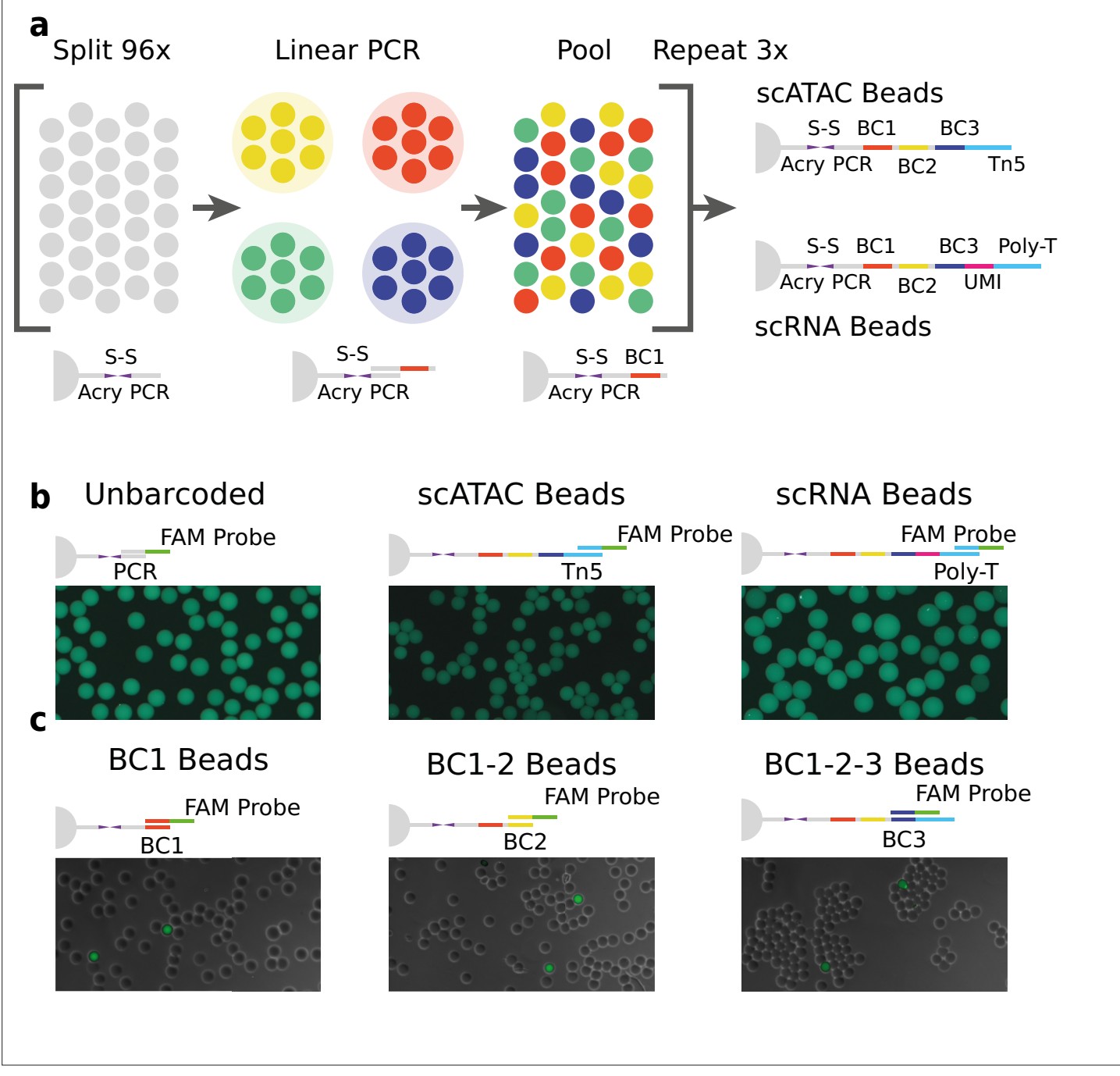

**Figure 1.** Technical overview of HyDrop barcoded bead production and quality control. (**a**) Split-pool process for barcoding of dissolvable hydrogel beads for single-cell RNA-sequencing (scRNA-seq) and single-cell assay for transposase-accessible chromatin (scATAC-seq). Unbarcoded hydrogel beads are sequentially distributed over 96 wells, sub-barcoded using linear amplification polymerase chain reaction (PCR), repooled, and re-distributed two more times to generate 96 × 96 × 96 (884,736) possible barcode combinations. Different 3-prime terminal capture sequences are possible depending on the oligonucleotide sequence appended in the last step. (**b**) Semiquantitative assessment of bead primer incorporation by fluorescence in-situ hybridisation (FISH) using a fluorescein amidite (FAM) probe after every sub-barcoding step shows that bead fluorescence uniformity is retained throughout the barcoding process. (**c**) FISH with FAM probes complementary to only one of 96 sub-barcode possibilities shows that approximately 1/96 beads exhibit fluorescence for a selected sub-barcode probe. Fluorescence signal is overlaid with a brightfield image at 50% transparency to indicate positions of nonfluorescent beads (see *Figure 1—figure supplements 2–6* for additional quality control experiments and full images).

The online version of this article includes the following source data and figure supplement(s) for figure 1:

**Figure supplement 1.** Hydrogel bead integrity after being frozen and thawed.

*Figure 1 continued on next page*

*Figure 1 continued*

**Figure supplement 2.** Fluorescence signal of freshly barcoded beads.

**Figure supplement 2—source data 1.** Count, mean diameter, and mean intensity of linear amplification and isothermal extension beads.

**Figure supplement 3.** Brightfield images of barcoded beads incubated with FAM oligonucleotide probe complementary to one out of 96 possible sub-barcodes.

**Figure supplement 3—source data 1.** Total counts and fluorescence-positive counts of beads incubated with fluoresceine amidite (FAM) oligonucleotide probe complementary to 1 out of 96 barcode possibilities.

**Figure supplement 4.** Fluorescence signal of nondissolvable 50 uM HyDrop-RNA beads.

**Figure supplement 4—source data 1.** Count, mean diameter, and mean background-adjusted intensity of beads incubated in 50 mM of DTT for 15 and 50 min as well as negative control.

**Figure supplement 5.** HyDrop-ATAC emulsion before and after PCR, with dissolvable and non-dissolvable barcoded hydrogel beads.

**Figure supplement 6.** HyDrop-RNA bead fluorescence intensity and barnyard plots for acrydite primer concentrations of 50, 12 and 6 uM.

---

*supplement 4*, Table S3), dissolvable beads did not disrupt the emulsion during the thermocycling (*Figure 1—figure supplement 5*). We hypothesize that this effect is due to the lower physical stress generated by the dissolved hydrogel as opposed to the solid hydrogel. This added benefit allowed us to implement scATAC-seq (see further) on HyDrop. Finally, by varying the concentration of the acrydite primer during bead synthesis, lower or higher amounts of cleavable barcoded primers could be generated. We found that when the concentration of acrydite primer incorporated in the hydrogel matrix was high (50 µM, similar to InDrop), excess unreacted barcodes could not be sufficiently filtered out in further downstream steps. These primers were then carried over to subsequent reactions, leading to random barcoding of free fragments after droplet merging, and subsequently to cell-mixed expression or chromatin accessibility profiles. The bead primer concentration with an optimal balance between sensitivity and library purity was found to be 12 µM for both scATAC-seq and scRNA-seq (*Figure 1—figure supplement 6*).

## Implementation and accuracy assessment of HyDrop-ATAC

We implemented a new open-source protocol for single-cell ATAC-seq using HyDrop's dissolvable barcoded hydrogel beads. Nuclei were Tn5 tagmented in bulk and coencapsulated with HyDrop-ATAC beads. Pitstop, a selective small molecule clathrin inhibitor, was supplemented during nuclei extraction and tagmentation to increase nucleus permeability to Tn5 (*Mulqueen et al., 2019*). Inside the droplet, the hydrogel beads dissolve and release their uniquely barcoded primers inside the droplet due to the presence of DTT carried by the nuclei/PCR mix. Subsequent thermocycling of the emulsion denatures the Tn5 protein complex and releases accessible chromatin fragments within the droplet. These fragments were then linearly amplified and cell indexed by the bead's barcoded primers after which the emulsion was broken and the indexed ATAC fragments were pooled, PCR amplified, and sequenced (*Figure 2*, see – *Supplementary file 2* 'Molecular sequence description of HyDrop-ATAC' for an in-depth visualization of the nucleic acid sequences in every step).

In order to coencapsulate beads and nuclei with a high capture rate and minimal microfluidic complexity, we developed a custom microfluidic chip (*Figure 3a and b*). The chip design features one inlet for beads, one inlet for cells or nuclei, and one inlet for the emulsion carrier oil. This configuration is slightly more convenient to operate compared to inDrop's four channel setup. Several layers of passive filters near the inlet ports mitigate dust and debris buildup during droplet generation to prevent obstruction of the channels. Beads and nuclei were loaded via a tip reservoir to reduce nonlinear flow behavior and the potential accumulation of cells/nuclei and hydrogel beads associated with narrow tubes (*Sinha et al., 2019*; *Hur et al., 2011*; *Figure 3b*). Due to the stability of all flows and the deformable nature of the hydrogel beads, >90% occupancy of hydrogel beads in droplets could be achieved (*Abate et al., 2009*; *Figure 3c*), resulting in a final cell recovery of ~65%.

To assess the purity of scATAC-seq libraries generated by HyDrop-ATAC, we performed two mixed species experiments. First, we generated single-nuclei ATAC-seq libraries from a 50:50 mixture of human breast cancer (MCF-7) and a mouse melanoma cell line generated previously (*Dankort et al., 2009*). We developed a custom preprocessing and mapping pipeline for HyDrop-ATAC data (*Dankort*

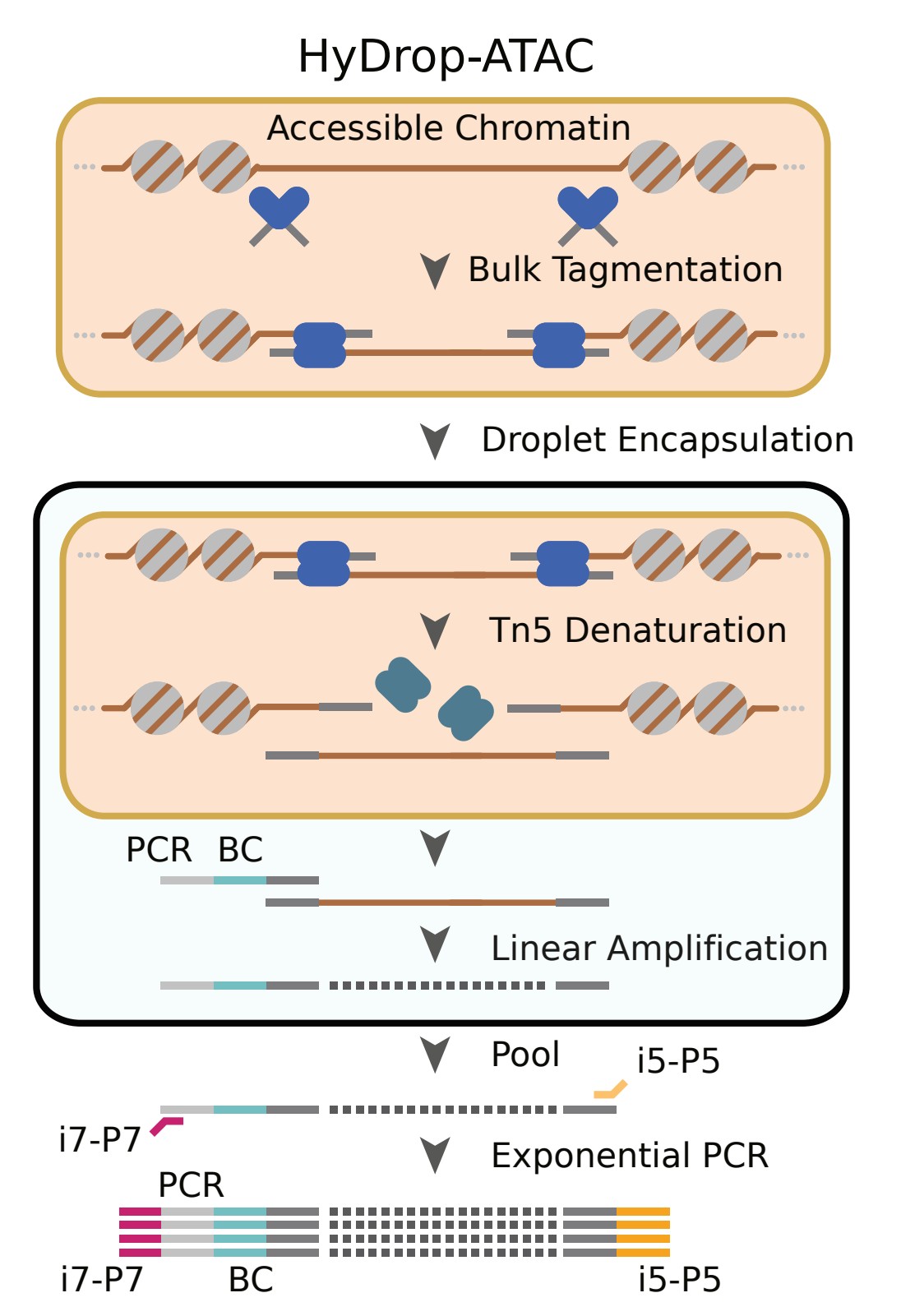

**Figure 2.** Schematic overview of HyDrop-ATAC. Nuclear membrane is visualized in salmon, water droplet is visualized in blue. Nuclei were Tn5 tagmented in bulk and co-encapsulated with HyDrop-ATAC beads, where the hydrogel beads dissolve and release their uniquely barcoded primers. Thermocycling of the emulsion releases accessible chromatin fragments which are then linearly amplified and cell indexed by the bead's barcoded primers within the droplet. The emulsion is then broken and the indexed ATAC fragments are pooled, PCR amplified, and sequenced.

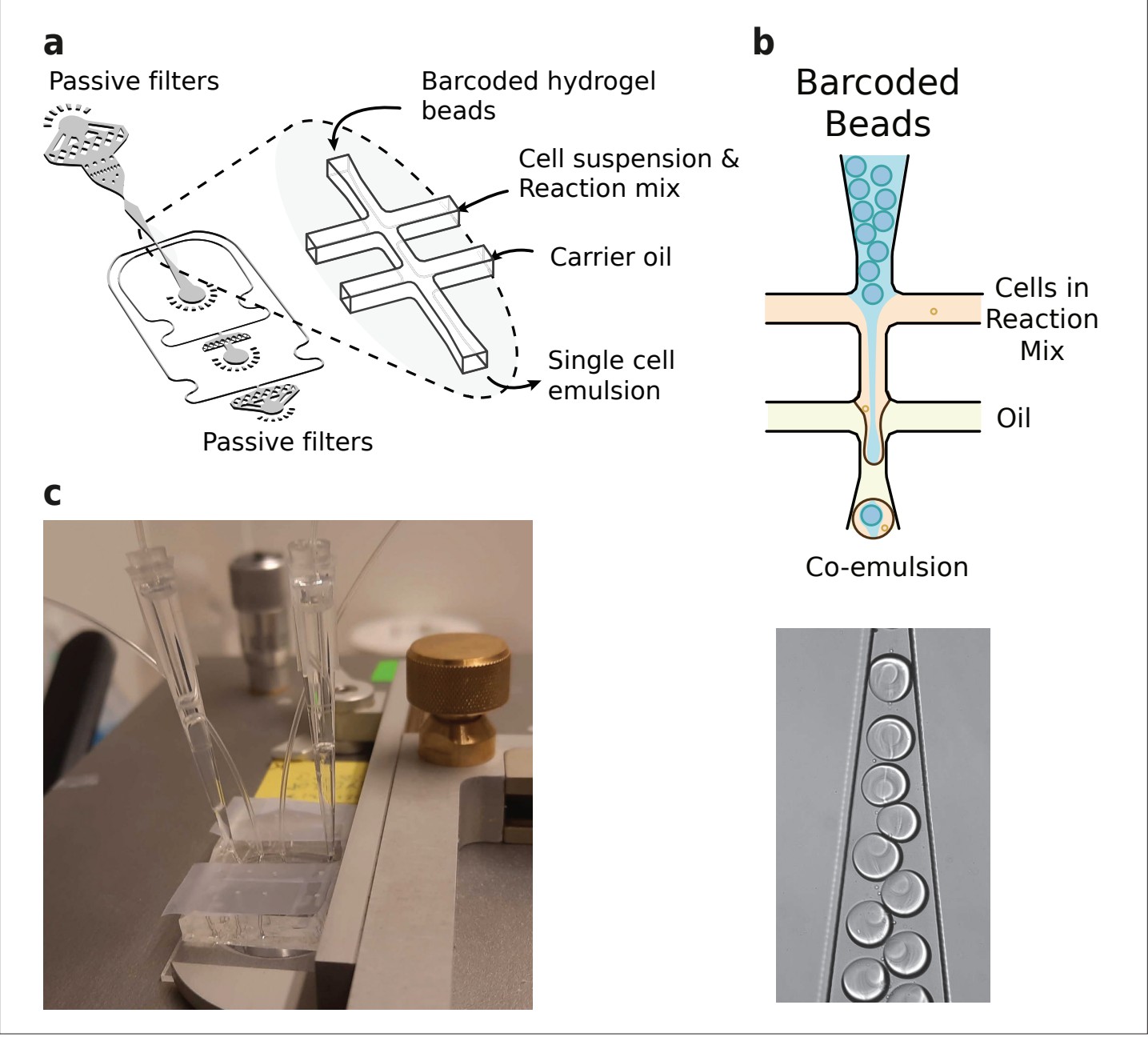

**Figure 3.** Microfluidics setup employed in HyDrop emulsion generation. (**a**) 3d rendering of HyDrop chip design. The design has three inlets: one each for carrier oil, barcoded hydrogel beads and cell/reaction mix (see **Figure 3—figure supplement 1** for full design). Passive filters at each inlet prevent dust and debris from entering the droplet generating junction. (**b**) Diagram and snapshot of cell/bead encapsulation into microdroplets. (**c**) Microfluidic chip setup on the Onyx integrated microfluidic instrument. Cells and beads are loaded into pipette tips and plugged into a HyDrop chip. Flow of oil and aqueous phases is achieved by Onyx displacement syringe pumps.

The online version of this article includes the following figure supplement(s) for figure 3:

**Figure supplement 1.** Microfluidic chip designs for cell/bead encapsulation and hydrogel bead generation - oil, monomer mix, cell suspensions, and beads can be pumped in through their respective inlets.

*et al., 2009*). After filtering the cell barcodes for a minimum transcription start sites (TSS) enrichment score of 7 and unique fragment count of 1000, we recovered 1353 cells from an input of 2000, and a median of 2705 unique fragments per cell. About 88% of all barcode reads were paired with the whitelist when one mismatch was allowed along the entire length of the 30 bp barcode. An additional 1.16% and 1.13% of HyDrop barcodes could be assigned to the whitelist when two and three

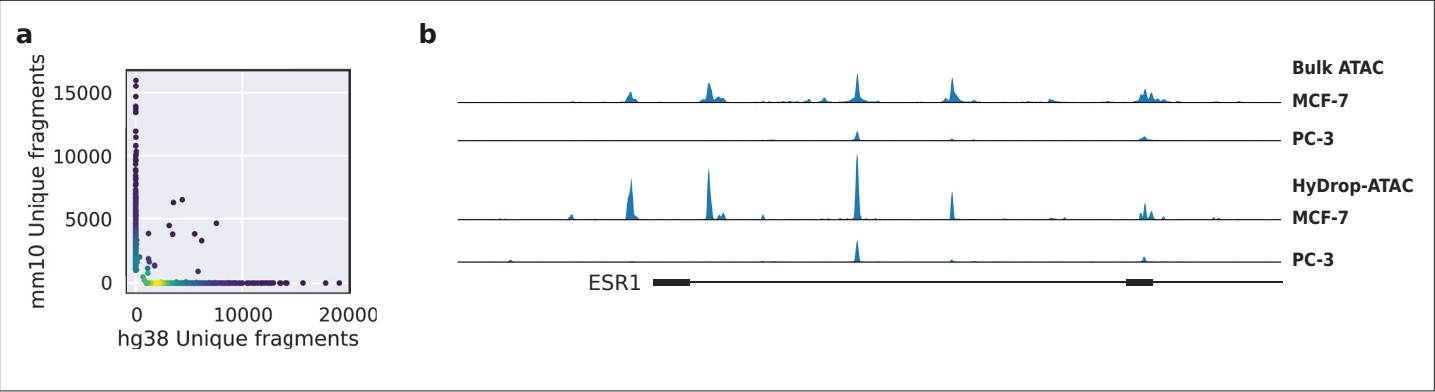

**Figure 4.** Validation of HyDrop-ATAC on mixed species cell lines. (**a**) Scatterplot of the number of unique fragments detected in a 50:50 mixture of human Michigan Cancer Foundation-7 (MCF-7) and mouse melanoma cells colored by local density estimation. (**b**) reads per genome coverage (RPGC)-normalized aggregate genome tracks comparing HyDrop-ATAC and bulk ATAC-seq profiles of human MCF-7 and prostate cancer-3 (PC-3) cell lines around the estrogen receptor 1 (ESR1) locus, scaled to maximum of all four samples. Aggregate enrichment profile of reads around transcription start site, see *Figure 4—figure supplement 1* for clustering of these cells and count correlation with public data. Supplementary source data files available for figure (a).

The online version of this article includes the following figure supplement(s) for figure 4:

**Figure supplement 1.** UMAP and count correlations within bulk ATAC-seq data for MCF-7 and PC-3 HyDrop-ATAC.

mismatches were allowed, respectively. The remaining 9% of barcode sequences could not be identified and are most likely a result of frameshift due to incorporation errors in the synthetic oligonucleotide production process. Notably, only 39.7% of barcode reads could be matched with the whitelist in a public inDrop dataset (*Kalish et al., 2020*), when one mismatch was allowed. When prefiltered cell barcode sequences were used as a whitelist for a public mouse retina Drop-seq dataset (*Macosko et al., 2015*), 49.5% of reads could be matched. We identified 98.4% of cells as either human or mouse at a minimum purity of 95% fragments mapping to either species (*Figure 4a*). Next, we generated libraries from a mixture of MCF-7/PC-3/Mouse cortex (45:45:10) to evaluate whether two human cell types can be distinguished. A spike-in of 10%mouse cells was used as an internal control. We recovered 2602 human cells, 466 mouse cells, and 93 species doublets after filtering for 95% species purity. Clustering human cells (together with the MCF-7 cells from the first species mixing experiment to evaluate batch effects) recovered two distinct populations, each exhibiting specific ATAC-seq peaks near MCF-7 or PC-3 marker genes (*Figure 4—figure supplement 1a*). Aggregated reads per cluster showed typical ATAC-seq profiles concordant with public bulk ATAC-seq data (*Consortium, 2012*; *Figure 4b*, *Figure 4—figure supplement 1*).

## Application of HyDrop-ATAC on flash-frozen mouse cortex recapitulates cerebral cellular heterogeneity and cell-type specific accessibility profiles

To evaluate the performance of HyDrop-ATAC on primary tissue, we generated single-cell libraries from snap-frozen, dissected adult mouse brain cortex. Libraries were sequenced to approximately 75% duplication rate. After filtering for a minimum of 1000 unique nuclear fragments, a TSS enrichment score of 5, and removing 506 cells (6%) detected as doublets by Scrublet (*Wolock et al., 2019*), we recovered a total of 7996 single nuclei. Cells passing the filters had a median of 4148 fragments per cell, a median TSS enrichment score of 13, and a median of 53% of fragments in peaks, reflecting high-quality cells and low levels of background signal (*Figure 5a–d*). Even though the number of unique fragments per cell (~4 K) is lower than that of commercial methods (e.g. 17–20 K per cell for 10× Genomics, see Methods), HyDrop-ATAC achieves comparable results in terms of TSS enrichment and fraction of reads in peaks scores. We used cisTopic (*Bravo González-Blas et al., 2019*) to reduce the dimensionality of the dataset and the Leiden algorithm (*Traag et al., 2019*) to cluster cells (*Figure 5e*). Cell annotation using the aggregated ATAC signal around several neocortex markers (*Yao et al., 2021*; *Zeisel et al., 2018*) recovered 19 distinct cell types, similar to previously published scATAC-seq mouse cortex data (*Preissl et al., 2018*; *Li et al., 2020*; *Figure 5—figure supplement*

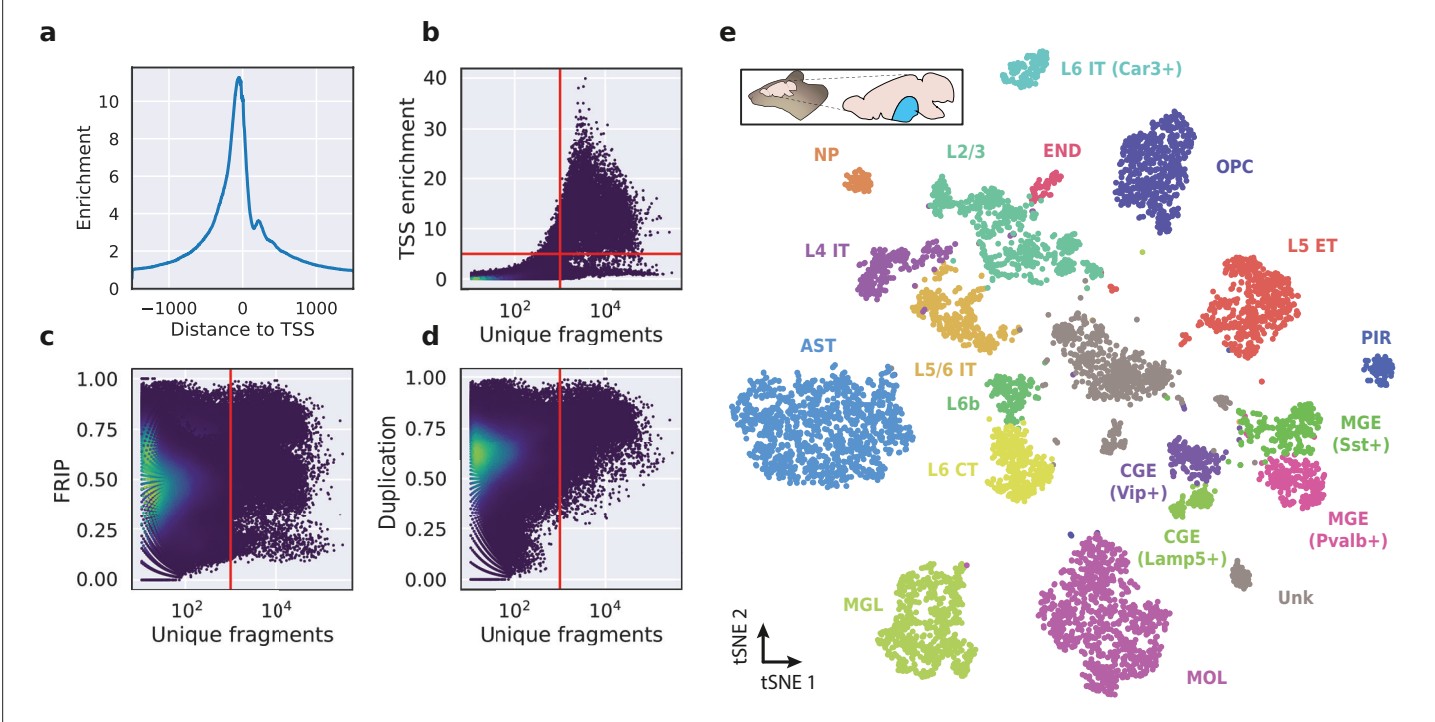

**Figure 5.** Application of HyDrop-ATAC on flash-frozen mouse cortex recapitulates cellular heterogeneity. Aggregate enrichment of ATAC fragments near transcription start sites (TSS) (**a**), TSS enrichment per barcode (**b**), fraction of reads in peaks (FRIP) per barcode, (**c**) and duplication rate per barcode (**d**) in mouse cortex HyDrop-ATAC data. A minimum TSS enrichment of 5 and a unique number of fragments of 1000 are used as cut-off values to separate cells from background (red lines). Cells are colored by local density estimation. (**e**) UMAP projection of 7996 mouse cortex nuclei annotated with cell type inferred by accessibility near marker genes. Abbreviations: microglia (MGL), mature oligodendrocytes (MOL), oligodendrocyte precursors (OPC), astrocytes (AST), endothelial cells (END), piriform cortex neurons (PIR), caudal and medial ganglionic eminence derived neurons (CGE, MGE), layers 2–6 intratelencephalic (IT), L5 extratelencephalic (ET), L5/6 near projecting excitatory neurons (NP), L6 corticoencephalic (CT), and deep L6 excitatory neurons (L6b). See *Figure 5—figure supplement 1* for cluster marker gene activities. Supplementary source data files available for figures b, c, and d.

The online version of this article includes the following figure supplement(s) for figure 5:

**Figure supplement 1.** Heatmap of mouse cortex HyDrop-ATAC gene activity – gene activity was imputed by normalized accessibility within a 10 kb window around the gene.

*1*). For example, we identified oligodendrocyte precursors and mature oligodendrocytes, marked by exclusive accessibility nearby *Sox10* and *Pdgfra2*, respectively. Within ganglionic eminence-derived interneurons, we were able to further distinguish medial ganglionic eminence-derived subtypes with specific ATAC-seq signal near either *Vip* or *Lamp5*, and caudal ganglionic eminence-derived subtypes with accessibility near either *Sst* or *Pvalb*. Finally, HyDrop-ATAC data revealed distinct cell-type specific differentially accessible regions (*Figure 6a and b*).

## Implementation of HyDrop-RNA as a hybrid method between inDrop and Drop-Seq

We next implemented a new scRNA-seq assay using barcoded bead primers carrying a 3-prime poly (dT) sequence. Single cells or nuclei were resuspended in a reverse transcriptase mix and coencapsulated into microdroplets with 3-prime poly (dT) HyDrop beads. The same microfluidic chip design was used for both HyDrop-RNA and HyDrop-ATAC. Cells are lysed inside the droplets upon contact with the lysis buffer in which the barcoded beads were suspended. Simultaneously, barcoded primers were released from the hydrogel bead after exposure to DTT present in the reverse transcriptase mix. Reverse transcription inside the emulsion generates thousands of barcoded single-cell cDNA libraries in parallel. The emulsion was then broken and the single-cell transcriptome libraries were processed further in a pooled manner (*Figure 7*, see *Supplementary file 3*).

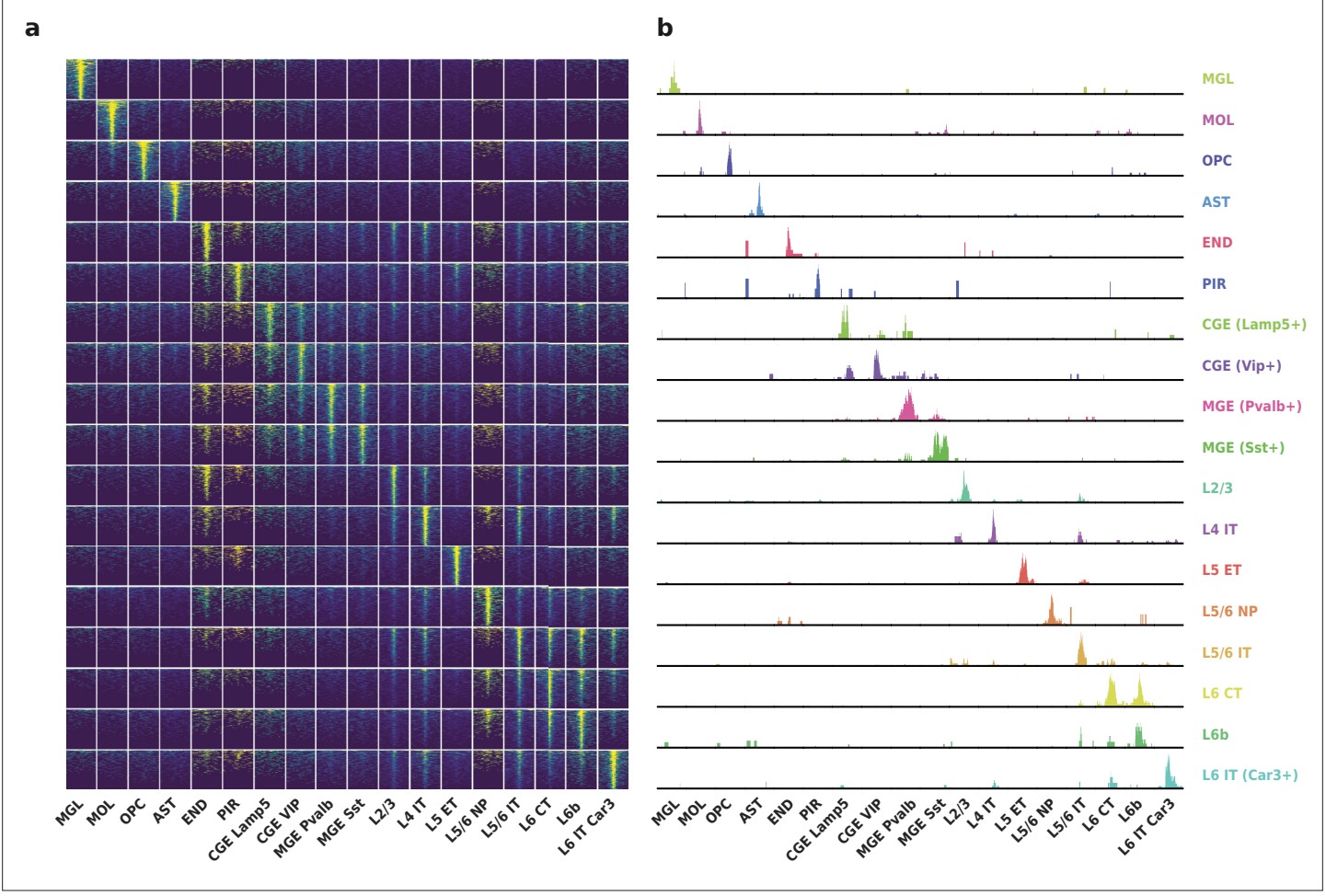

**Figure 6.** Differentially accessible regions (DAR) between cell types recovered by HyDrop-ATAC on mouse cerebral cortex. (**a**) Aggregate accessibility of top 1000 DAR per cluster. (**b**) Row-scaled, counts-per-million (CPM)-normalized aggregate genome track covering the top one DAR for each cluster.

'Molecular sequence description of HyDrop-RNA' for an in-depth visualization of the nucleic acid sequences in every step, similarly to the InDrop protocol (*Zilionis et al., 2017*). Although both assays are based on hydrogel beads, HyDrop-RNA differs significantly from InDrop. HyDrop-RNA employs a template switching oligo (TSO) reverse transcription technique (similar to Drop-seq), rather than an in vitro transcription/random hexamer priming workflow. This change simplifies and speeds up the protocol significantly with no reduction in sensitivity.

To improve the assay's sensitivity, we interrogated the impact of several alterations to the protocol's reaction chemistry by testing them on a 50:50 human-mouse (human melanoma, mouse melanoma) mixture. We first investigated the use of pooled exonuclease I treatment after reverse transcription to remove unused barcode primers. We reasoned that these unused barcoded primers could potentially prime transcripts during the subsequent bulk inverse-suppressive polymerase chain reaction (IS-PCR), leading to a loss of purity of transcripts associated with a given barcode. As evident from the increase in pure cells detected, we found that exonuclease I treatment indeed improved assay purity (*Figure 8—figure supplement 1*). We then tested the implementation of a locked nucleic acid (LNA) in the TSO, as it has been shown to increase assay sensitivity due to increased stability of the TSO-cDNA complex (*Picelli et al., 2014*). We also investigated the addition of guanosine triphosphate (GTP) and polyethylene glycol (PEG) to the in-droplet reverse transcription reaction. Both the addition of PEG as a molecular crowding agent and GTP as a stabilizer of the TSO complex has been shown to improve assay sensitivity in SMART-seq3 (*Hagemann-Jensen et al., 2020*). Finally, we wondered whether a second strand synthesis library construction approach could outperform the TSO/IS-PCR approach. In order to test this, we compared both alkaline hydrolysis and enzymatic treatment (RNAse H) to

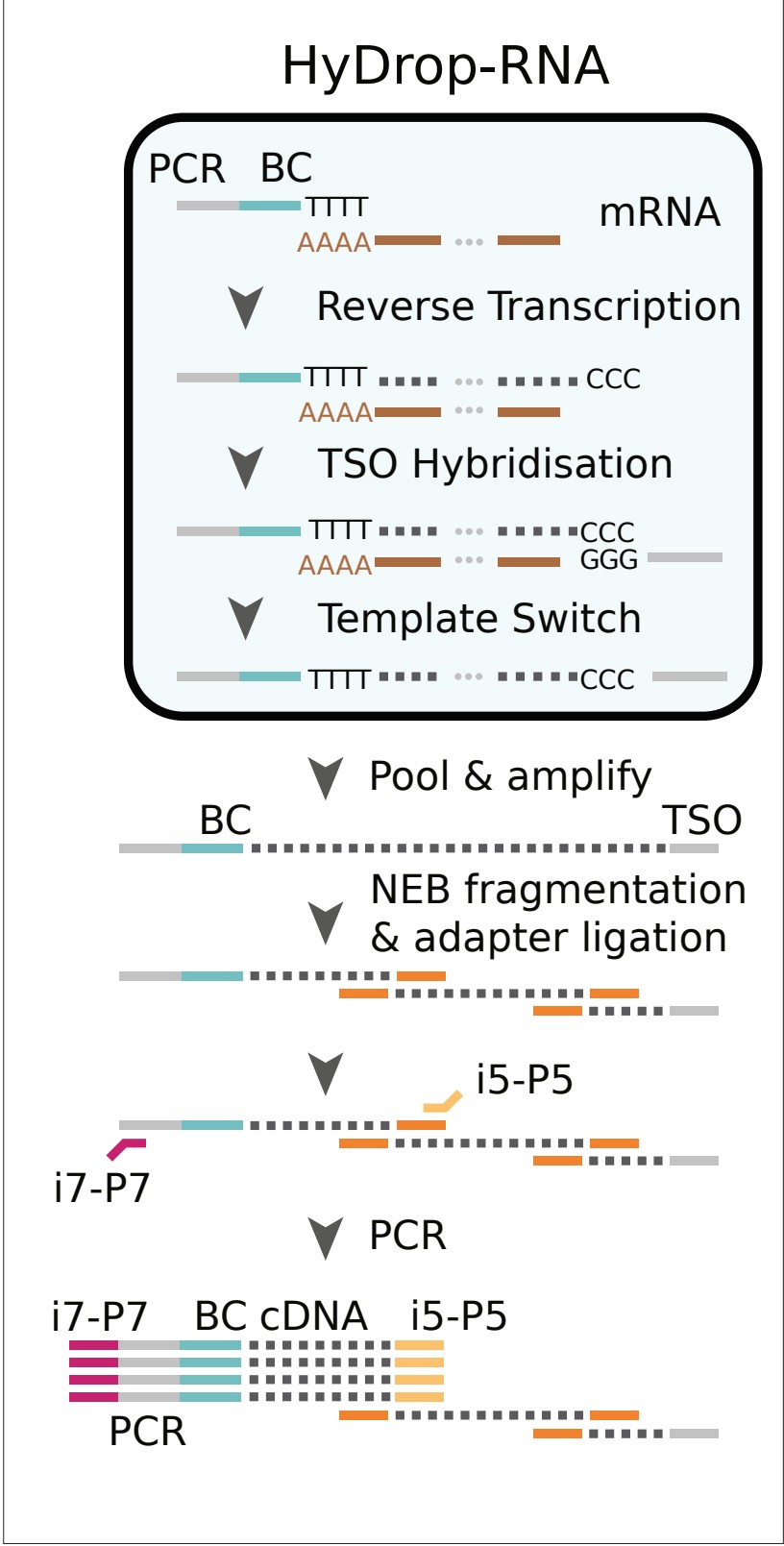

**Figure 7.** Schematic overview of HyDrop-ATAC. Single cells or nuclei are resuspended in a reverse transcriptase mix and coencapsulated into microdroplets with 3-prime poly (dT) HyDrop beads. Cells are lysed inside the droplets and barcoded primers (BC) are released from the hydrogel bead. Reverse transcription inside the emulsion using a template-switching oligo (TSO) generates thousands of barcoded single-cell cDNA libraries in parallel which are processed further in a pooled manner after breaking the emulsion.

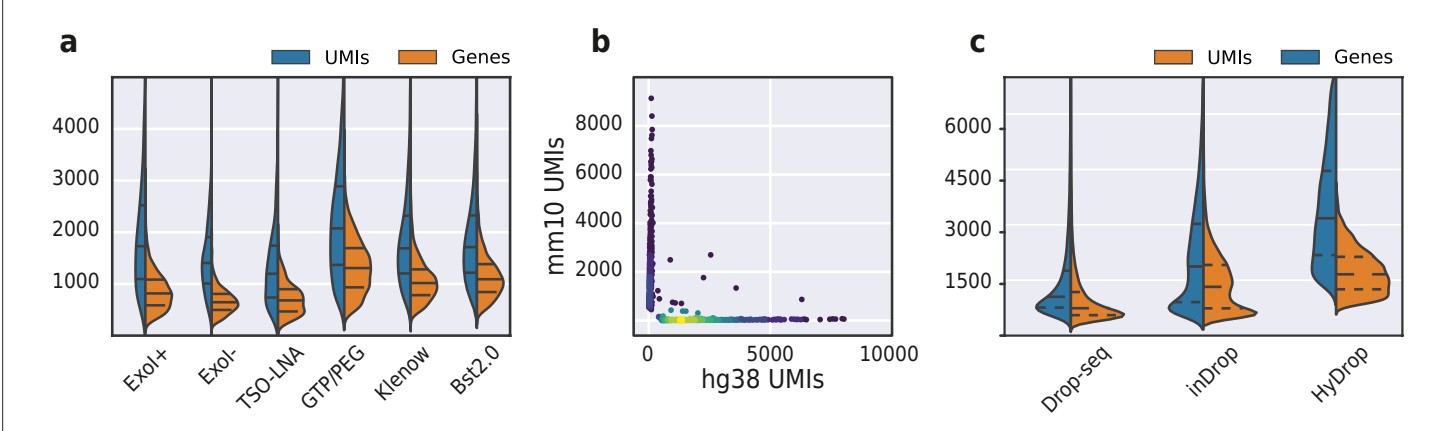

**Figure 8.** Validation and benchmarking of HyDrop-RNA on species mixed cell line samples and mouse cerebral cortex. (**a**) Comparison of unique molecular identifier (UMI) and gene count of HyDrop-RNA with and without Exo I treatment postdroplet merging, with the use of a locked nucleic acid (LNA) template switching oligo (TSO) and with GTP/PEG, BST2.0, and Klenow fragment library preparation. Inner lines represent Q1, median, and Q3. See *Figure 8—figure supplement 1* for species purity plots of these experiments. (**b**) Scatterplot of human and mouse UMIs detected in a 50:50 mixture of human MCF-7 and mouse melanoma cells colored by local density estimation. (**c**) Comparison of UMI and gene count of public inDrop mouse cortex data, public Drop-seq mouse retina data, and HyDrop-RNA mouse cortex data. Inner lines represent Q1, median, and Q3. See *Figure 8— figure supplement 2* for additional quality comparison including commercial methods. Supplementary source data files available for figures a, b, and c.

The online version of this article includes the following figure supplement(s) for figure 8:

**Figure supplement 1.** HyDrop-RNA species-mixing purity plots for several different protocol versions.

**Figure supplement 2.** Collection of quality control metrics for inDrop, Drop-seq, HyDrop-RNA, and 10× datasets on mouse brain cells.

remove the RNA strand from the first strand product, and evaluated the performance of both Bst 2.0 DNA polymerase and Klenow (exo-) (*Hughes et al., 2020*; *Stickels et al., 2021*) fragment for second strand synthesis. We found that the classical TSO and IS-PCR protocol supplemented with GTP/PEG performed best, yielding a median of 2110 UMIs and 1325 genes per cell with a species purity of 90.1% (*Figure 8a* and *Figure 8—figure supplement 1*). Accordingly, the GTP/PEG method was used in all further HyDrop-RNA experiments. Applying this protocol to a 50:50 human-mouse (MCF-7, mouse melanoma) mixture recovered 1235 human and 846 mouse cells with 169 species doublets at a cutoff of 95% species purity, with a median of 1439 UMIs and 904 genes per cell (*Figure 8b*).

## HyDrop-RNA on flash-frozen mouse cortex recovers cerebral cell types

We then used HyDrop-RNA to generate 9508 single-nuclei transcriptomes from snap-frozen mouse cortex in a single experiment. At a sequencing saturation of approximately 60% duplicates, we achieve a median of 3389 UMIs and 1658 genes per cell before, and 3404 UMIs and 1662 genes per cell after doublet filtering (*Figure 8c*), compared to the median of 1920 UMIs and 1321 genes detected by inDrop snRNA-seq on mouse auditory cortex neurons (*Kalish et al., 2020*) and the median of 1071 UMIs and 763 genes detected by Drop-seq on mouse retina neurons (*Macosko et al., 2015*). Both datasets also exhibited a lower read alignment efficiency compared to HyDrop, with Drop-seq mapping 52%/21%, and inDrop mapping 48%/28% of reads to the mouse genome/transcriptome, while HyDrop reaches 88%/55%. HyDrop's 'sequencing efficiency' is higher than both inDrop and Drop-seq's: 7.44% of HyDrop's sequenced reads end up as mapped transcripts with a UMI, whereas this metric is 3.24% for the inDrop dataset and 4.46% for the Drop-seq dataset (*Figure 8—figure supplement 2*). 10× Chromium v2 gene expression reference data reports a median number of genes of 775–2679 and a median number of UMIs of 1127–6570 on E18 and adult mouse brain nuclei, and a sequencing efficiency of 2.54%–20.9% (see methods). Comparing the top percluster differentially expressed genes with markers from the Allen Brain Atlas SMART-seq data (*Yao et al., 2021*) revealed 30 distinct populations corresponding to previously identified cell types (*Figure 9a-d*, *Figure 9— figure supplement 1*). In addition to the major neuronal and glial populations previously detected in our HyDrop-ATAC experiment, we detect a small population of vascular leptomeningeal cells and layer two intratelencephalic neurons from the medial entorhinal area (L2 IT ENTm). We also detect both D1 and D2 medium spiny neurons as a result of residual striatal tissue and layer 3 *Scnn1a* + neurons from

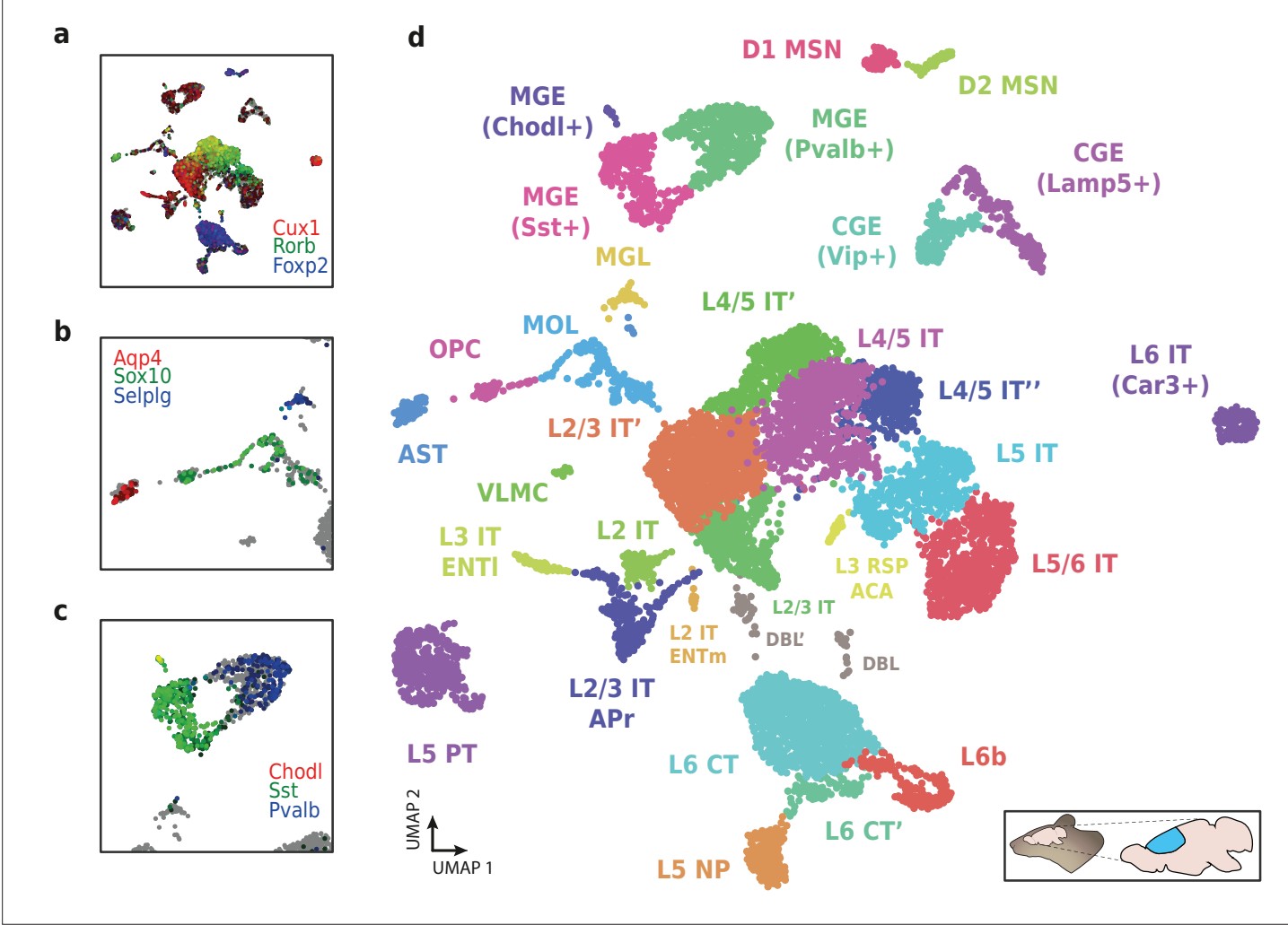

**Figure 9.** Application of HyDrop-RNA on flash-frozen mouse cerebral cortex. Mouse cortex UMAP colored by log-scaled unique molecular identifier counts of Cux1, Rorb, Foxp2 (**a**), Aqp4, Sox10, Selplg (**b**), Chodl, Sst, and Pvalb (**c**). Colors are scaled to minimum and maximum values. (**d**) UMAP projection of 9507 mouse cortex nuclei annotated with cell type inferred by marker gene expression. Abbreviations: microglia (MGL), mature oligodendrocytes (MOL), oligodendrocyte precursors (OPC), astrocytes (AST), endothelial cells (END), piriform cortex neurons (PIR), caudal and medial ganglionic eminence derived neurons (CGE, MGE), layers 2–6 intratelencephalic (IT), pyramidal tract (PT), near projecting excitatory neurons (NP) and corticoencephalic (CT) neurons, layer two intratelencephalic medial entorhinal area neurons (L2 IT ENTm), L2/3 intratelencephalic area prostriata neurons (L2/3 IT APr), layer three intratelencephalic entorhinal neurons (L3 IT ENTl), layer retrosplenial and anterior cingulate area neurons (L3 RSP ACA), deep L6 excitatory neurons (L6b), D1 and D2 medium spiny neurons (MSN), and vascular leptomeningeal cells (VLMC). See *Figure 9—figure supplement 1* for expression of top differentially accessible genes within these clusters.

The online version of this article includes the following figure supplement(s) for figure 9:

**Figure supplement 1.** Expression of HyDrop-RNA mouse cortex top three differentially expressed genes from each cluster.

the retrosplenial and anterior cingulate area (L3 RSP ACA), concordant with Atlas SMART-seq data (*Yao et al., 2021*).

## HyDrop-RNA on FAC-sorted *Drosophila* neurons confirms high capture rate even on small input samples

To assess HyDrop-RNA's performance on low cell input samples, we also performed the protocol on approximately 1500 FAC-sorted neurons from the *Drosophila* brain. We dissected brains in which mCherry expression was driven in specific cell populations by a Gal4 driver line (R74G01-Gal4) and used mCherry-positive cells as input for HyDrop-RNA (*Figure 10a*). Of the 1500 cells obtained after FACS sorting, we recovered 973 fly brain cells with a median of 1307 UMIs and 640 genes

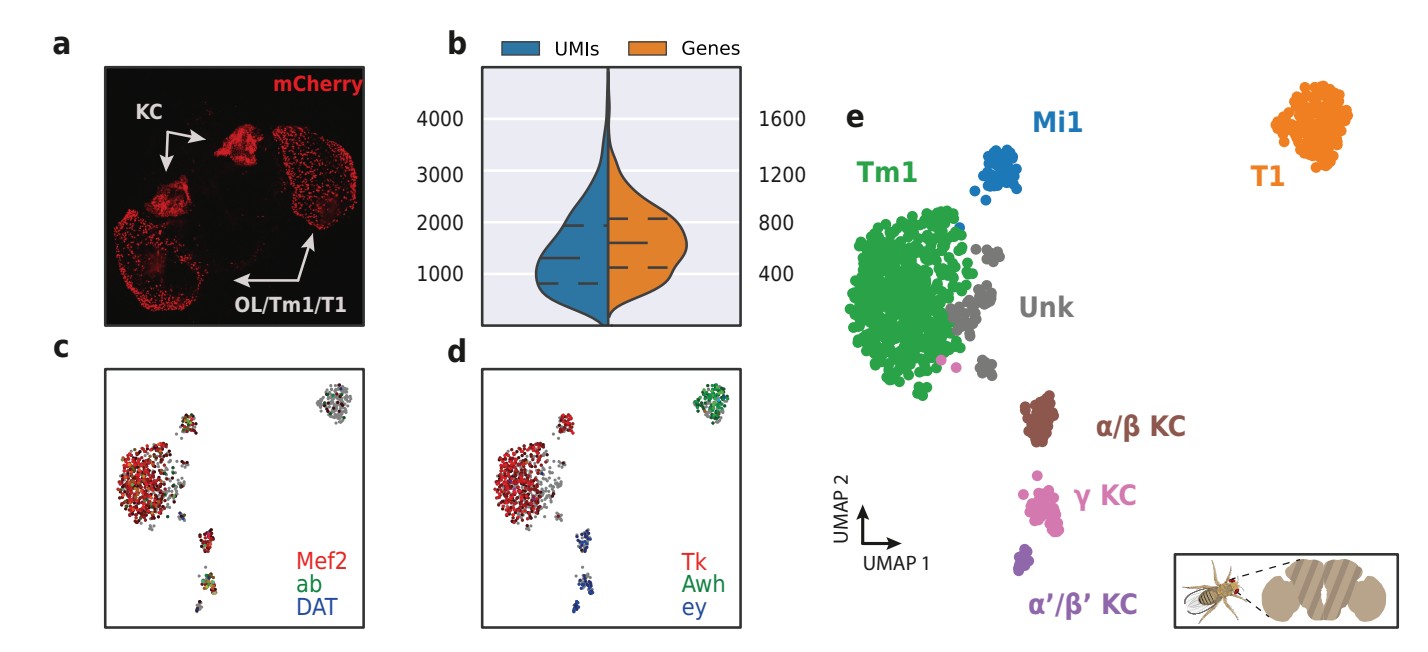

**Figure 10.** Application of HyDrop-RNA on FAC-sorted *Drosophila* neurons. (**a**) Confocal maximum intensity projection of R74G01-Gal4> UAS-mCherry brain. (**b**) Violin plot of unique molecular identifiers (UMIs) and genes detected in nuclei derived from FAC-sorted fly neurons. Inner lines represent Q1, median, and Q3. Fly neuron UMAP colored by log-scaled UMI counts of Mef2, ab, DAT (**c**) and Tk, Awh, ey (**d**). Colors are scaled to minimum and maximum values. (**e**) UMAP of 973 FAC-sorted *Drosophila* neurons annotated with cell types inferred by marker gene expression (KC, Kenyon cells; Tm1, transmedullary neuron; Mi1, medullary intrinsic neuron). See *Figure 10—figure supplement 1* for expression of marker genes within these clusters. Supplementary source data file available for figure b.

The online version of this article includes the following figure supplement(s) for figure 10:

**Figure supplement 1.** Expression of marker genes of HyDrop-RNA on FAC-sorted fly brain cells.

(*Figure 10b*). In comparison, in-house Drop-seq performed on the entire fly brain recovered a median of 579 UMIs and 289 genes per cell at a deeper sequencing saturation (*Davie et al., 2018*). Annotation of the 973 single-cell transcriptomes obtained by HyDrop confirmed the presence of all three Kenyon cell subtypes alongside T1 and Tm1 neurons, as expected from our stainings and previous reports (*Konstantinides et al., 2018*). Surprisingly, we also detected a small population of Mi1 neurons (*Figure 10c-e*, *Figure 10—figure supplement 1*).

## Discussion

We developed HyDrop, an open-source platform for single-cell RNA- or single-cell ATAC-seq using hydrogel beads. We applied HyDrop to generate thousands of mouse, human, and *Drosophila* single-cell gene expression and chromatin accessibility profiles to demonstrate the protocol's applicability to a variety of different biological samples. HyDrop-ATAC and HyDrop-RNA experiments on mouse and fly tissues recapitulated the cellular heterogeneity of these complex samples and strongly agreed with reference datasets (*Davie et al., 2018*; *Yao et al., 2021*; *Preissl et al., 2018*; *Li et al., 2020*). We found HyDrop-RNA to outperform its open-source predecessors (*Klein et al., 2015*; *Macosko et al., 2015*; *Chen et al., 2019*) both in terms of sensitivity and user-friendliness. Moreover, at a per-cell library cost of <\$0.03 (*Supplementary file 5*) it does so at a significantly lower cost compared to commercial droplet-microfluidic alternatives (*Zheng et al., 2017*; *Satpathy et al., 2019*; *Lareau et al., 2019*).

Optimization and modification of HyDrop's reaction chemistry and bead composition may lead to further improvements in sensitivity. For example, we consider that benchmarking additional bead barcoding strategies, such as direct on-bead DNA synthesis or barcode ligation (*Delley and Abate, 2020*) instead of linear amplification may further improve overall barcode quality. Ligation strategies would also allow the incorporation of oligonucleotide modifications in the barcode's capture sequence,

a possible necessity for adaptation of HyDrop to novel assays. For example, this change would permit HyDrop's extension to protocols such as s3-ATAC (*Mulqueen et al., 2021*), which leverages the use of uracil bases and uracil-intolerant polymerases to double the amount of usable fragments generated by Tn5. The current 96 *Kalish et al., 2020* design could also be changed to 384 *Macosko et al., 2015* (like inDrop) or even 768 *Macosko et al., 2015*. This two-step barcoding approach would result in shorter barcode sequences, allowing for cost-saving in sequencing and increased flexibility when pooling HyDrop libraries together with other library types such as 10× Genomics 3' gene expression libraries. However, rescaling the barcodes in this manner would also necessitate the acquisition of a larger pool of sub-barcodes - from three 96 well plates to 8 and 16 plates respectively - and reduce the total barcode complexity from 884,736 to 147,456 and 589,824. Finally, one of the main challenges of implementing custom microfluidics systems is the considerable price of accurate stepper motor syringe pumps (~$2000 per independent channel) and the need for trained personnel. However, several simplified alternatives have recently emerged (*Langer et al., 2018*; *Kim et al., 2016*), and we believe their application with HyDrop could further reduce the start-up costs and complexity of single-cell experiments.

We envision that HyDrop's reduction in both cost and labor will accelerate the scaling of large-scale atlasing efforts and bring the benefits of single-cell sequencing to smaller projects. Additionally, HyDrop's dissolvable bead synthesis and linear amplification barcoding toolkit could potentially be exploited to produce more complex beads incorporating multiple capture sequences, leading to implementation of single-cell (multi) omics assays such as the capture of accessible chromatin, (m)RNA, and surface epitopes (*Stoeckius et al., 2017*) or intracellular proteins (*Rivello et al., 2020*) from single cells.

## Materials and methods
### Microfluidic droplet generator manufacturing

Microfluidic droplet generators were produced using standard SU-8 lithography and polydimethylsiloxane (PDMS) lithography according to well-established protocols (*Xia and Whitesides, 1998*). Briefly, the designs for droplet generators were made in AutoCAD R2014 and the designs are printed onto a chrome mask using a laser writer. The SU-8 lithography is performed on a 4-inch silicon wafer using SU 8–2050 (Microchem) negative photoresist using UV aligner (EVG-620). As per manufacturers' recommendation, spin coating of the wafer with SU8 was performed at 500 rpm (ramp 100 rpm/s) - 10 s and 2000 rpm (ramp: 300 rpm/s) - 30 s, to achieve a feature height of 70–80 µm. For preparing the PDMS chip, a mixture of PDMS monomer and cross-linker (Dow Corning SYLGARD 184) was prepared at a ratio of 10:1 and mixed thoroughly. The mixture was degassed in vacuum for 45 min and poured onto an SU-8 master template and baked at 80°C for 4 hr. Inlet ports were cut using a 1-mm biopsy needle after which the chips were exposed to high-voltage plasma for 30 s and bonded onto a glass slide. 5 µL of 2% Trichloro (1 H,1H,2H,2H-perfluorooctyl)silane in hydrofluoroether (HFE) was injected into each channel, incubated for 10 min at room temperature and excess oil was removed by applying pressurized air. Chips were finally baked at 100°C for 2 hr (more detailed methods for photolithography and PDMS lithography in https://doi.org/10.17504/protocols.io.bvpin5ke).

### Barcoded hydrogel bead manufacturing and storage

Dissolvable hydrogel beads are synthesized similar to a previously published protocol *Ren et al., 2021* and barcoded according to a modified inDrop protocol (*Zilionis et al., 2017*). For synthesizing 2–3 mL batch of beads, 2 mL of Bead Monomer Mix (6% acrylamide, 0.55% bisacryloylcystoylamine, 10% Tris-buffered saline with EDTA and Triton X-100 (TBSET) (10 mM Tris-HCl pH 8, 137 mM NaCl, 2.7 mM KCl, 10 mM EDTA, 0.1% Triton X-100), 12 µM acrydite primer, 0.6% ammonium persulfate) was encapsulated into 50 µm diameter droplets in HFE-7500 Novac oil with EA-008 surfactant (RAN Biotech). 1 mL aliquots of the resulting emulsion was layered with 400 µL of mineral oil and incubated at 65°C for 14 hr. Excess mineral oil and the emulsion oil were removed and 2–3 washes with 1 mL of droplet breaking solution (20% PFO in HFE) were performed. Beads were pelleted at 5000 xg, 4°C for 30 s and washed twice in 1 mL of 1% SPAN-80 in hexane. Beads were sequentially washed in TBSET until all hexane phase was removed.

Beads were washed twice in Bead Wash Buffer (10 mM Tris-HCl pH 8, 0.1% Tween-20), twice in PCR Wash Buffer (10 mM Tris-HCl pH8, 50 mM KCl, 1.5 mM MgCl2, 0.1% Tween-20). The subsequent liquid handling in the 96-well plate is performed using Hamilton microlab STAR robot. 22.5 µL of beads were distributed to a 96-well plate. 2.5 µL of 100 µM sub-barcode primer and 25 µL of KAPA HiFi Hotstart master mix (Roche) were added to each well and the plate was thermocycled (95°C 3 min, 5 cycles of 98°C 20 s, 38°C 4 min, 72°C 2 min, 1 cycle of 98°C 1 min, 38°C 10 min, 72°C 4 min, followed by a final hold on 4°C) with intermittent vortexing during every annealing step. 50 µL of STOP-25 (10 mM Tris-HCl pH 8, 25 mM EDTA, 0.1% Tween-20, 100 mM KCl) was added to each well to deactivate the polymerase and its contents were pooled. Remaining beads in wells were washed out with 100 µL of STOP-25 and the beads were rotated at room temperature for 30 min. Beads were then washed with STOP-10 (10 mM Tris-HCl pH 8, 10 mM EDTA, 0.1% Tween-20, 100 mM KCl) and rotated for 10 min in denaturation solution (150 mM NaOH, 85 mM BRIJ-35). Beads were washed twice in denaturation solution and twice more in neutralization solution (100 mM Tris-HCl pH 8, 10 mM EDTA, 0.1% Tween-20, 100 mM NaCl). The sub-barcoding step was repeated twice more for a total of three sub-barcodes.

Hydrogel beads were sequentially filtered using a 70 µm strainer (Falcon). For both the HyDrop-ATAC and RNA beads were pelleted at 300 xg, 4°C and resuspended in 5 mL of Bead Freezing Buffer (150 mM NaCl, 125 mM Tris-HCl pH 7, 10 mM MgCl2, 4% Tween-20, 0.75% Triton X-100, 30% glycerol, 0.3% BSA). Beads were pelleted at 300 xg, 4°C and resuspended in 5 mL of Bead Freezing Buffer a second time and incubated at 4°C for at least 3 hr. Beads were pelleted at 1000 xg, 4°C and the pellet was aliquoted into 35 µL aliquots and stored at –80°C for long-term storage (further method details in https://doi.org/10.17504/protocols.io.b4cyqsxw).

## Hydrogel bead fluorescence in situ hybridization quality control

Bead QC was performed as described previously (*Zilionis et al., 2017*). Briefly, 10 µL of hydrogel beads were resuspended in 1 mL of hybridization buffer (5 mM Tris-HCl pH 8.0, 5 mM EDTA, 0.05% Tween-20, 1 M KCl) and centrifuged for 1 min at 1000 xg. The wash was repeated once more, and 960 µL of the supernatant was removed. 4 µL of 200 µM specific FAM probe was added depending on which part of the barcode needed testing (see supplementary oligonucleotide table). The beads were incubated at room temperature in the dark for 30 min. Beads were washed thrice in QC buffer and visualized under a Zeiss Axioplan two microscope and a Zeiss Colibri seven light source (300 ms exposure time, 80% light source intensity). Occasional punctate intensities (diameter ~1 µm) were observed in quality control performed on 10× beads and are thought to be dye precipitates or crystals.

Fluorescence images were analyzed using opencv2 (*Bradski, 2015*). Briefly, foreground and background were determined using Otsu's method. Circles were detected in the foreground using Hough's algorithm. The mean grayscale intensity of the background was subtracted from the grayscale intensity at the center of each detected circle. The center intensity method was favored over a volumetric intensity measurement over the entire bead as slight differences in focus between images can affect intensity near the edge of the bead. For the 1/96 sub-barcode experiment, both fluorescence and brightfield images were taken of the same beads incubated with 1 out of 96 possible sub-barcodes. Total number of beads was counted manually from the brightfield image, while positive beads were counted using the image analysis tool from the fluorescence images. A mask of the positive beads from the fluorescence image was overlaid onto the brightfield image to mark positive beads for visualization. Image analysis tool code and raw data of all used images available at https://github.com/aertslab/hydrop_data_analysis, (copy archieved at swh:1:rev:059bf5a7779dc2894670ecf5f820c-14bceb68493; *Rop, 2022* and https://doi.org/10.5281/zenodo.6415968).

## Cell culture and cell dissociation

MCF-7 cells (NCI-DTP Cat# MCF7, RRID:CVCL_0031) were cultured in RPMI1640 (ThermoFisher 11875093) medium supplemented with 10% FBS (ThermoFisher 10270–106), 1% penicillin/streptomycin (Life Technologies 15140122), and 10 ug/mL insulin (Sigma Aldrich I9278) and passaged twice per week. PC-3 cells (NCI-DTP Cat# PC-3, RRID:CVCL_0035) were cultured in RPMI1640 medium supplemented with 10% FBS and 1% penicillin/streptomycin and passaged twice per week. Mouse melanoma cells were cultured in DMEM (ThermoFisher 13345364) supplemented with 10% FBS and 1% penicillin/streptomycin and passaged once per week. MM087 melanoma cells were cultured in F-10 Nutrient mix supplemented with 10% FBS and 1% penicillin/streptomycin and passaged once

per week. All cell lines tested negative for mycoplasma prior to use. The identity of PC-3, MCF-7, and mouse melanoma cell lines were correctly identified since their aggregate scATAC-seq data correlated highly with public data (see *Figure 4b*, *Figure 4—figure supplement 1* for PC-3 and MCF-7, mouse melanoma data not shown). The identity of our MM087 was confirmed by SNP comparison with the MM087 genome for our manuscript 'Robust gene expression programs underlie recurrent cell states and phenotype switching in melanoma' (Wouters et al., https://doi.org/10.1038/s41556-020-0547-3).

 Cells were washed in PBS and dissociated into single-cell suspensions by adding 1.5 mL of 0.05% trypsin (Life Technologies 25300054) and waiting for 5–7 min. The single-cell suspension was centrifuged at 500 rcf for 5 min at 4°C and the resulting pellet was resuspended in PBS. This PBS wash was repeated once more and the single-cell suspension was processed further.

## Fly rearing and cell dissociation

GMR74G01-Gal4 (BDSC Cat# 39868, RRID:BDSC_39868) and UAS-mCherry (BL#38425) flies were obtained from Bloomington *Drosophila* Stock Center. The resulting cross-strain was raised on standard cornmeal-agar medium at 25°C at a 12 hr light/dark cycle. Fifty adult brains were dissected in DPBS and transferred to a tube containing 100 μL of cold DPBS solution. Samples were centrifuged at 500 rcf for 1 min and the supernatant was replaced by 50 μL of dispase (3 mg/mL, Sigma-Aldrich, D4818, 2 mg) and 75 μL of collagenase I (100 mg/mL, Invitrogen, 17100–017). Brains were dissociated in a Thermoshaker (Grant Bio PCMT) at 500 rpm for 2 hr at 25°C, with pipette mixing every 15 min. Cells were subsequently washed with 1000 μL of cold DPBS solution and centrifuged at 500 rcf for 5 min at 4°C and resuspended in 250 μL of DPBS with 0.04% BSA. Cell suspensions were passed through a 10 μM pluriStrainer (ImTec Diagnostics, 435001050). Cells were sorted based on viability and mCherry positivity using the Sony MA900 cell sorter. Sorted cells were collected into Eppendorf tubes precoated with 1% BSA.

## Cell line nuclei extraction

A pellet of one million dissociated cells or fewer was incubated on ice in 200 μL of ATAC lysis buffer (1% BSA, 10 mM Tris-HCl pH 7.5, 10 mM NaCl, 0.1% Tween-20, 0.1% NP-40, 3 mM MgCl$_2$, 70 μM Pitstop in DMSO, 0.01% digitonin) for 5–7 min. 1 mL of ATAC nuclei wash buffer (1% BSA, 10 mM Tris-HCl pH 7.5, 0.1% Tween-20, 10 mM NaCl, 3 mM MgCl$_2$) was added and the nuclei were pelleted at 500 xg, 4°C for 5 min. The resulting pellet was resuspended in 100 μL of ice-cold PBS and filtered with a 40 μm strainer (Flowmi).

## Mouse cortex dissection

All animal experiments were conducted according to the KU Leuven ethical guidelines and approved by the KU Leuven Ethical Committee for Animal Experimentation (approved protocol numbers ECD P183/2017). Mice were maintained in a specific pathogen-free facility under standard housing conditions with continuous access to food and water. Mice used in the study were 57 days old and were maintained on 14 hr light, 10 hr dark light cycle from 7 to 21 hr. In this study, cortical brain tissue from female P57 BL/6Jax was used. Animals were anesthetized with isoflurane, and decapitated. Cortices were collected, divided in four equal quadrants along the dorsoventral and anterior-posterior axis, and immediately snap frozen in liquid nitrogen. For HyDrop-ATAC, a ventral/posterior quadrant from the left hemisphere was used. For HyDrop-RNA, a dorsal/anterior quadrant was used from the left hemisphere of a second mouse.

## Snap-frozen mouse cortex nuclei extraction

For the preparation of nuclei for scRNA-seq, we used a modified protocol from the recently published single-nuclei preparation toolbox *Delley and Abate, 2020* to extract nuclei from snap-frozen mouse cortex samples. Briefly, a ~1 cm$^3$ frozen piece of mouse cortex tissue was transferred to 0.5 mL of ice-cold homogenization buffer (Salt-tris solution - 146 mM NaCl, 10 mM Tris 7.5, 1 mM CaCl$_2$, 21 mM MgCl$_2$, 250 mM sucrose, 0.03% Tween-20, 0.01% BSA, 25 mM KCl, 1 mM 2-mercaptoethanol, 1× complete protease inhibitor, 0.5 U/ul of RNAse In Plus (Promega)) in a Dounce homogenizer mortar and thawed for 2 min. The tissue was homogenized with 10 strokes of pestle A and 5 strokes of pestle B until a homogeneous nuclei suspension was achieved. The resulting homogenate was filtered through a 70-μm cell strainer (Corning). The homogenizer and the filter are rinsed with an additional

500 µL of homogenization buffer. The tissue material was pelleted at 500 xg and the supernatant was discarded. The tissue pellet was resuspended in a homogenization buffer without Tween-20. An addition 1.65 ml of homogenization buffer was topped up and mixed with 2.65 ml of gradient medium (75 mM sucrose, 1 mM CaCl$_2$, 50% Optiprep, 5 mM MgCl$_2$, 10 mM Tris 7.5, 1 mM 2-mercaptoethanol 1× complete protease inhibitor, 0.5 U/ul of RNAse In Plus (Promega)). 4 mL of 29% iodoxanol cushion was prepared with a diluent medium (250 mM sucrose, 150 mM KCl, 30 mM MgCl$_2$, 60 mM Tris 8) and was loaded into an ultracentrifuge tube. 5.3 mL of sample in homogenization buffer + gradient medium was gently layered on top of the 29% iodoxanol cushion. Sample was centrifuged at 7700 xg, 4°C for 30 min and the supernatant was gently removed without disturbing the nuclei pellet. Nuclei were resuspended in 100 µL of nuclei buffer (1% BSA in PBS + 1 U/ul of RNAse Inhibitor).

For the preparation of nuclei for ATAC-seq, we used a slightly modified protocol to extract nuclei from snap-frozen mouse cortex samples. Briefly, a ~1 cm$^3$ frozen piece of mouse cortex tissue was transferred to 1 mL of ice-cold homogenization buffer (320 mM sucrose, 10 mM NaCl, 3 mM Mg (OAc), 10 mM Tris 7.5, 0.1 mM EDTA, 0.1% IGEPAL-CA360, 1× complete protease inhibitor and 1 mM DTT) in a Dounce homogenizer mortar and thawed for 2 min. The tissue was homogenized with 10 strokes of pestle A and 5 strokes of pestle B until a homogeneous nuclei suspension was achieved. The resulting homogenate was filtered through a 70 µm cell strainer (Corning). 2.65 mL of ice-cold gradient medium was added to 2.65 mL of homogenate and mixed well. 4 mL of 29% iodoxanol cushion (129.2 mM sucrose, 77.5 mM KCl, 15.5 mM MgCl, 31 mM Tris-HCl pH 7.5, 29% iodoxanol) was loaded into ultracentrifuge tube. 5.3 mL of sample in homogenization buffer + gradient medium was gently layered on top of the 29% iodoxanol cushion. Sample was centrifuged at 7700 xg, 4°C for 30 min and the supernatant was gently removed without disturbing the nuclei pellet. Nuclei were resuspended in 100 µL of nuclei buffer (1% BSA in PBS).

## HyDrop-ATAC library preparation

A total of 50,000 nuclei were resuspended in 50 µL of ATAC reaction mix (10% DMF, 10% Tris-HCl pH 7.4, 5 mM MgCl$_2$, 5 ng/µL Tn5, 70 µM Pitstop in DMSO, 0.1% Tween-20, 0.01% digitonin) and incubated at 37°C for 1 hr. 100 µL of 0.1% BSA in PBS was added and the nuclei were pelleted at 500 xg, 4°C for 5 min and resuspended in 40 µL of 0.1% BSA in PBS.

Tagmented nuclei were added to 100 µL of PCR mix (1.3× Phusion HF Buffer, 15% Optiprep, 1.3 mM dNTPs, 39 mM DTT, 0.065 U/µL Phusion HF polymerase, 0.065 U/µL Deep Vent polymerase, 0.013 U/µL ET SSB). PCR mix was coencapsulated with 35 µL of freshly thawed HyDrop-ATAC beads in HFE-7500 Novac oil with EA-008 surfactant (RAN Biotech) on the Onyx microfluidics platform (Droplet Genomics). The resulting emulsion was collected in aliquots of 50 µL total volume and thermocycled according to the linear amplification program (72°C 15 min, 98°C 3 min, 13 amplification cycles of [98°C 10 s, 63°C 30 s, 72°C 1 min], followed by a final hold on 4°C). 125 µL of recovery agent (20% PFO in HFE), 55 µL of GITC buffer (5 M GITC, 25 mM EDTA, 50 mM Tris-HCl pH 7.4) and 5 µL of 1 M DTT was added to each separate aliquot of 50 µL thermocycled emulsion and incubated on ice for 5 min. 5 µL of Dynabeads was added to the aqueous phase and incubated for 10 min. Dynabeads were pelleted on a Nd magnet and washed twice with 80% EtOH. Elution was performed in 50 µL of EB-DTT-Tween (10 mM DTT, 0.1% Tween-20 in EB (10 mM Tris-HCl, pH 8.5)). A 1× Ampure bead purification was performed according to manufacturer's recommendations. Elution was performed in 30 µL of EB-DTT (10 mM DTT in EB). Eluted library was further amplified in 100 µL of PCR mix (1× KAPA HiFi, 1 µM index i7 primer, 1 µM index i5 primer). Final library was purified in a 0.4×–1.2× double-sided Ampure purification and eluted in 25 µL of EB-DTT (10 mM DTT in EB). User-friendly protocol available on https://www.protocols.io/view/hydrop-atac-v1-0-bxsbpnan. A detailed cost analysis of HyDrop-ATAC can be found in supplementary sheet 2.

## HyDrop-RNA single-cell library preparation

For a recovery of 2000 cells, 3795 cells were resuspended in 85 µL of RT mix (1× Maxima RT Buffer, 0.9 mM dNTPs, 25 mM DTT, 1.3 mM GTP, 15% Optiprep, 1.3 U/µL RNAse inhibitor, 15 U/µL Maxima hRT, 12.5 µM TSO, 4.4% PEG-8000). RT mix was co-encapsulated with 35 µL of freshly thawed HyDrop-RNA beads in RAN oil on the Onyx microfluidics platform. The resulting emulsion was collected in aliquots of 50 µL total volume and thermocycled according to the RT program (42°C for 90 min, 11 cycles of [50°C for 2 min, 42°C for 2 min], 85°C for 5 min, followed by a final hold on

4°C). 125 µL of recovery agent (20% PFO in HFE), 55 µL of GITC Buffer (5 M GITC, 25 mM EDTA, 50 mM Tris-HCl pH 7.4) and 5 µL of 1 M DTT was added to each separate aliquot of 50 µL thermocycled emulsion and incubated on ice for 5 min. 99 µL of Ampure XP beads was added to the aqueous phase and incubated for 10 min. Ampure beads were pelleted on a Nd magnet and washed twice with 80% EtOH. Elution was performed in 30 µL of EB-DTT-Tween (10 mM DTT, 0.1% Tween-20 in EB). Exonuclease treatment was performed by adding 4 µL of 10× NEBuffer 3.1, 4 µL of Exo I, and 2 µL of dH2O to 30 µL of eluted library. The Exo I reaction mix was incubated at 37°C for 5 min, 80°C for 1 min for heat inactivation followed by a final hold at 4°C. 2 µL of 1 M DTT was added and a 0.8× Ampure XP purification was performed according to manufacturer's recommendations. cDNA was eluted in 40.5 µL of EB-DTT (10 mM DTT in EB) and added to ISPCR mix (40 µL library, 50 µL 2× KAPA HiFi, 10 µL 10 µM TSO-P primer). PCR cycling was performed according to the ISPCR program (95°C for 3 min, 13 cycles of [98°C for 20 s, 63°C for 20 s, 72°C for 3 min.], 72 °C for 5 min) followed by a final hold at 4°C. 2 µL of 1 M DTT was added and a 0.6× Ampure XP purification was performed according to manufacturer's recommendations. cDNA was eluted in 28.5 µL of EB-DTT. Final sequencing library was prepared according to the following customized NEB Ultra II FS protocol (NEB E7805S). 80 ng of amplified cDNA was fragmented in Ultra II fragmentation mix (26 µL of amplified cDNA, 7 µL of NEBNext Ultra II FS Reaction Buffer, 2 µL of NEBNext Ultra II FS Enzyme Mix) on the following thermocycling program: 37°C for 10 min, 65°C for 30 min, and a final hold at 4°C. 15 µL of EB was added and a 0.8× Ampure purification was performed according to manufacturer's recommendation and eluted in 35 µL. Fragmented library was adapter ligated in NEBNext Ultra II adapter ligation mix (35 µL of fragmented library, 30 µL of NEBNext Ultra II Ligation Master Mix, 1 µL of NEBNext Ligation Enhancer, 2.5 µL of NEBNext Adapter for Illumina) at 20°C for 15 min, with 4°C final hold. 28.5 µL of EB was added and a 0.8× Ampure purification was performed according to manufacturer's recommendation and eluted in 30 µL. Eluted library was amplified in PCR master mix (50 µL 2× KAPA HiFi, 10 µL 10 µM Hy-i7 primer, 10 µL 10 µM Hy-i5 primer, 30 µL eluted library) in the following thermocycling program: 95°C for 3 min, 13 cycles of (98°C for 20 s, 64°C for 30 s, 72°C for 30 s), 72°C for 5 min, and a final hold at 4°C. Sequencing-ready library was purified using a 0.8× Ampure purification and eluted in 30 µL of EB. User-friendly protocol available on https://doi.org/10.17504/protocols.io.b4xwqxpe.

## HyDrop-RNA optimization trials

We performed six trials on a 50:50 mixture of human melanoma (MM087) and mouse melanoma (MMel). Trials were performed as described in the general HyDrop-RNA protocol, but with the following changes. All trials, except for the GTP/PEG trial, were performed using the following RT reaction mix (1.6× Maxima h-RT buffer, 1.6 mM dNTPs, 47 mM DTT, 15% Optiprep, 1.6 U/µL RNAse Inhibitor, 15.7 U/µL Maxima hRT, 12.5 µM TSO). For the exocondition, the exonuclease I treatment was skipped. For all other conditions the exonuclease I treatment was performed as described above. For the TSO-LNA trial, an LNA TSO was used instead of the regular TSO. For the GTP/PEG trial, all steps were performed as described in the main protocol.

For the Klenow fragment second strand synthesis trial, the purified first strand product was treated with 1 µL of *Escherichia coli* RNase H (NEB M0297S). The mixture was incubated at 37°C for 30 min after which the enzyme was inactivated using 10 mM EDTA. The single stranded product was purified using 1.2× Ampure XP bead purification (BD sciences) and eluted in 25 µL of EB buffer. dN-SMRT primer was added to the single strand product to a final concentration of 2.5 µM and the mixture was denatured by incubation at 95°C for 5 min. The sample was then allowed to cool to room temperature and incorporated in the Klenow enzyme mix (1× Maxima h-RT buffer, 1 mM dNTP, 1 U/µL of Klenow Exo-; NEB M0212L) was added to the single strand library. The Klenow enzyme mix was incubated at 37°C for 60 min. The second strand reaction was stopped by heating the product at 85°C for 5 min. The sample was purified using 1× Ampure XP and eluted in 40 µL of EB buffer. The purified second strand product was amplified with ISPCR primers as described above.

For the BST 2.0 polymerase second strand synthesis trial, the purified first strand product was treated with 1 µL of *E. coli* RNase H (NEB M0297S). The mixture was incubated at 37°C for 30 min after which the enzyme was inactivated using 10 mM EDTA. The single stranded product was purified using 1.2× Ampure XP bead purification (BD sciences) and eluted in 25 µL of EB buffer. dN-SMRT primer was added to the single strand product to a final concentration of 2.5 µM and the mixture was denatured by incubation at 95°C for 5 min. The sample was then allowed to cool to room temperature

and incorporated in the Bst 2.0 enzyme mix (1× Isothermal amplification buffer, 1 mM dNTP, 1 U/μL of Bst 2.0 DNA polymerase; NEB M0537L) was added to the denatured library and the mixture was incubated at 55°C for 10 min and 60°C for 45 min. The second strand reaction was stopped by heating the product at 85°C for 5 min. The sample was purified using 1× Ampure XP and eluted in 40 μL of EB buffer. The purified second strand product was amplified with ISPCR primers as described above.

## Sequencing

HyDrop-ATAC libraries were sequenced on Illumina NextSeq500 or NextSeq2000 systems using 50 cycles for read 1 (ATAC paired-end mate 1), 52 cycles for index 1 (barcode), 10 cycles for index 2 (sample index), and 50 cycles for read 2 (ATAC paired-end mate 2).

HyDrop-RNA libraries were sequenced on Illumina NextSeq2000 systems using 50 cycles for read 1 (3-prime cDNA), 10 cycles for index 1 (sample index, custom i7 read primer), 10 cycles for index 2 (sample index), and 58 cycles for read 2 (barcode+ UMI, custom read two primer).

## HyDrop-ATAC data processing

Barcode reads were trimmed to exclude the intersub-barcode linear amplification adapters using a mawk script. Next, the VSN scATAC-seq pre-processing pipeline (*Waegeneer et al., 2021*) was used to map the reads to the reference genome and generate a fragments file for downstream analysis. Here, barcode reads were compared to a whitelist (of 884,736 valid barcodes), and corrected, allowing for a maximum 1 bp mismatch. Uncorrected and corrected barcodes were appended to the fastq sequence identifier of the paired end ATAC-seq reads. Reads were mapped to the reference genome using bwa-mem with default settings, and the barcode information was added as tags to each read in the bam file. Duplicate-marking was performed using samtools markdup. In the final step of the pipeline, fragment files were generated using Sinto (https://github.com/timoast/sinto). For mixed species data, cells were filtered for a minimum of 1000 unique fragments and a minimum TSS enrichment of 7. For mouse cortex data, higher level analysis such as clustering and differential accessibility were performed using cisTopic (*Bravo González-Blas et al., 2019*). In brief, cells were filtered for a minimum of 1000 unique fragments and a minimum TSS enrichment of 5. Fragments overlapping mouse candidate cis-regulatory regions (*ENCODE Project Consortium et al., 2020*) were counted, and the resulting matrix was filtered for potential cell doublets using a Scrublet (*Wolock et al., 2019*) threshold of 0.35. Cells were Leiden clustered based on the cell-topic probability matrix generated by an initial cisTopic LDA incorporating 51 topics, at a resolution of 0.9 with 10 neighbours. A consensus peak set was generated from per-cluster peaks and used to recount fragments. Cells were filtered using the same filtering parameters and a new model with 50 topics was trained. Cells were again Leiden clustered based on the cell-topic probability matrix generated by the second LDA, at a resolution of 0.9 with 10 neighbours. Region accessibility was imputed based on binarized topic-region and cell-topic distributions. Gene activity was imputed based on Gini index-weighted imputed accessibility in a 10 kb up/downstream decaying window around each gene including promoters. Leiden clusters were annotated based on imputed gene accessibility around marker genes (*Yao et al., 2021*; *Zeisel et al., 2018*). Differentially accessible regions were called using one-versus-all Wilcoxon rank-sum tests for each cell type, with an adjusted p-value of 0.05 and $log_2$FC of 1.5. RPGC-normalized aggregate genome coverage bigwigs were generated from BAM files using DeepTools (*Ramírez et al., 2016*). Per-cluster genome coverage tracks were generated using pyBigWig.

## HyDrop-RNA data processing

Barcode reads were trimmed to exclude the intersub-barcode linear amplification adapters using a mawk script. Reads were then mapped and cell-demultiplexed using STARsolo (*Kaminow et al., 2021*) in CB_UMI_Complex mode. The resulting STARsolo-filtered count matrices were further analyzed using Scanpy (*Wolf et al., 2018*). In short, cells were filtered on expression of a maximum of 4000 genes, and a maximum of 1% UMIs from mitochondrial genes. Genes were filtered on expression in a minimum of three cells. Potential cell doublets were filtered out using a Scrublet (*Wolock et al., 2019*) threshold of 0.25. The filtered expression matrix was scaled to total counts and log-normalized. Total counts and mitochondrial reads were regressed out and UMAP embedding was performed after PCA. Cells were annotated and fine tuned based on differential gene expression of marker genes sourced

from either the Davie et al. *Drosophila* brain atlas (*Davie et al., 2018*) or the Allen Brain RNA-seq Database (*Yao et al., 2021*).

Raw inDrop (SRR10545068 to SRR10545079) and Drop-seq (SRR1853178 to SRR1853184) sequencing data was downloaded from SRA. Data was sampled to the HyDrop-RNA mouse cortex sample sequencing depth (52,738 reads per cell) and mapped to mouse reference genome using STARsolo, allowing one mismatch in the cell barcodes (like in our HyDrop-RNA and HyDrop-ATAC analyses). Full analysis process is documented on the Hydrop GitHub repository (https://github.com/ aertslab/hydrop_data_analysis/tree/main/HyDrop-RNA_publicdata_comparison). Public reference 10× single-cell ATAC-seq data was sourced from https://support.10xgenomics.com/single-cell-atac/ datasets ('Flash frozen cortex, hippocampus, and ventricular zone from embryonic mouse brain (E18)', Fresh cortex from adult mouse brain (P50)'). Public reference 10× single-cell gene expression data was sourced from https://support.10xgenomics.com/single-cell-gene-expression/datasets ('1 k Brain Nuclei from an E18 Mouse', '2 k Brain Nuclei from an Adult Mouse ( > 8 weeks)'). Public PC-3 and MCF-7 ATAC-seq data was sourced from ENCODE (ENCFF772EFK: doi:10.17989/ENCSR422SUG, ENCFF024FNF: doi:10.17989/ENCSR499ASS).

Data was visualized using a combination of Seaborn (*Waskom, 2021*) and Matplotlib (*Hunter, 2007*). A vector image representing mouse head and cortex was sourced from SciDraw (*Kennedy, 2020*).

## Acknowledgements

We thank Andrew Adey for his great advice on Tn5 and scATAC-seq. We also thank Sebastián Najle, Céline Vallot, and Arnau Sebé-Pedrós for many discussions on droplet microfluidics, Frederik Ceyssens and the KU Leuven Nanocenter for their support with microfabrication, Ghanem Ghanem for his kind donation of the MM087 melanoma lines, Jean-Christophe Marine for his kind donation of the mouse melanoma lines, and Koen De Wispelaere for his assistance during his master internship.

## Additional information

### Funding

| Funder | Grant reference number | Author |
|---|---|---|
| H2020 European Research Council | 724226_cis- CONTROL | Stein Aerts |
| KU Leuven | C14/18/092 | Stein Aerts |
| Fonds Wetenschappelijk Onderzoek | G0B5619N | Stein Aerts |
| Michael J. Fox Foundation for Parkinson's Research | ASAP-000430 | Christopher Campbell Flerin |
| Aligning Science Across Parkinson's | ASAP-000430 | Christopher Campbell Flerin |
| Foundation Against Cancer | 2016-070 | Stein Aerts |
| Stichting Tegen Kanker | | Jasper Wouters |
| Belgian Cancer Society | | Jasper Wouters |
| Fonds Wetenschappelijk Onderzoek | | Florian V De Rop Carmen Bravo González-Blas Jasper Janssens |
| VIB Tech Watch | | Suresh Poovathingal |

The funders had no role in study design, data collection and interpretation, or the decision to submit the work for publication.

## Author contributions

Florian V De Rop, Conceptualization, Data curation, Formal analysis, Funding acquisition, Investigation, Methodology, Validation, Visualization, Writing – original draft, Writing – review and editing; Joy N Ismail, Investigation, Writing – review and editing; Carmen Bravo González-Blas, Formal analysis, Software, Writing – review and editing; Gert J Hulselmans, Data curation, Software; Christopher Campbell Flerin, Software, Writing – original draft, Writing – review and editing; Jasper Janssens, Software; Koen Theunis, Methodology; Valerie M Christiaens, Gabriele Marcassa, Joris de Wit, Resources; Jasper Wouters, Writing – review and editing; Suresh Poovathingal, Conceptualization, Funding acquisition, Investigation, Methodology, Project administration, Supervision, Validation, Writing – original draft, Writing – review and editing; Stein Aerts, Conceptualization, Funding acquisition, Methodology, Project administration, Supervision, Writing – original draft, Writing – review and editing

## Author ORCIDs

Florian V De Rop ⓘ http://orcid.org/0000-0001-5241-924X
Jasper Wouters ⓘ http://orcid.org/0000-0002-7129-2990
Suresh Poovathingal ⓘ http://orcid.org/0000-0002-3236-2255
Stein Aerts ⓘ http://orcid.org/0000-0002-8006-0315

## Ethics

Ethical approvalAll animal experiments were conducted according to the KU Leuven ethical guidelines and approved by the KU Leuven Ethical Committee for Animal Experimentation (approved protocol numbers ECD P037/2016, P014/2017, and P062/2017). All use of cell lines was approved by the KU Leuven Ethical Committee for Research under project number S63316.

## Decision letter and Author response

Decision letter https://doi.org/10.7554/eLife.73971.sa1
Author response https://doi.org/10.7554/eLife.73971.sa2

# Additional files

## Supplementary files

• Supplementary file 1. "Molecular sequence description of HyDrop bead barcoding": visual description of progression of nucleotide sequences involved in every step of the barcoding process.

• Supplementary file 2. "Molecular sequence description of HyDrop-ATAC": visual description of progression of nucleotide sequences involved in every step of HyDrop-ATAC.

• Supplementary file 3. "Molecular sequence description of HyDrop-RNA": visual description of progression of nucleotide sequences involved in every step of HyDrop-RNA.

• Supplementary file 4. "Reagents and oligonucleotide list": list of all reagents and oligonucleotides used in HyDrop.

• Supplementary file 5. "Price calculation for HyDrop bead barcoding and HyDrop-ATAC": detailed price calculation for HyDrop bead barcoding and HyDrop-ATAC.

• Transparent reporting form

• Source data 1. Source data files for *Figure 4a*, *Figure 5b–d*, *Figure 8a–c*, *Figure 8—figure supplement 2*, and *Figure 10b*.

## Data availability

The data discussed in this publication have been deposited in NCBI's Gene Expression Omnibus and are accessible through GEO Series accession number GSE175684 (https://www.ncbi.nlm.nih.gov/geo/query/acc.cgi?acc=GSE175684) and on SCope (https://scope.aertslab.org/#/HyDrop/*/welcome). Source data files have been provided for Figures 2a, 3a, 3b, 3c and 3i, and can be regenerated data analysis tutorials for HyDrop. Data analysis tutorials for HyDrop are available on GitHub (https://github.com/aertslab/hydrop_data_analysis; copy archived at swh:1:rev:059bf5a7779dc2894670ecf5f820c14bceb68493).

The following dataset was generated:

| Author(s) | Year | Dataset title | Dataset URL | Database and Identifier |
|---|---|---|---|---|
| De Rop F, Ismail J, Bravo González-Blas C, Hulselmans G, Flerin C, Janssens J, Christiaens V, Poovathingal S, Aerts S | 2021 | HyDrop: droplet-based scATAC-seq and scRNA-seq using dissolvable hydrogel beads | https://www.ncbi.nlm.nih.gov/geo/query/acc.cgi?acc=GSE175684 | NCBI Gene Expression Omnibus, GSE175684 |

The following previously published datasets were used:

| Author(s) | Year | Dataset title | Dataset URL | Database and Identifier |
|---|---|---|---|---|
| Macosko EZ | 2015 | Drop-Seq analysis of P14 mouse retina single-cell suspension | https://www.ncbi.nlm.nih.gov/geo/query/acc.cgi?acc=GSE63472 | NCBI Gene Expression Omnibus, GSE63472 |
| Kalish BT, Greenberg ME | 2020 | inDrop scRNA-seq of mouse primary auditory cortex | https://www.ncbi.nlm.nih.gov/geo/query/acc.cgi?acc=GSE140883 | NCBI Gene Expression Omnibus, GSE140883 |
| 10x Genomics | 2019 | 10x scATAC-seq on flash frozen cortex, hippocampus, and ventricular zone from embryonic mouse brain (E18) | https://support.10xgenomics.com/single-cell-atac/datasets/1.2.0/atac_v1_E18_brain_flash_5k | 10x Genomics Resources, atac_v1_E18_brain_flash_5k |
| 10x Genomics | 2019 | 10x scATAC-seq on fresh cortex from adult mouse brain (P50) | https://support.10xgenomics.com/single-cell-atac/datasets/1.2.0/atac_v1_adult_brain_fresh_5k | 10x Genomics Resources, atac_v1_adult_brain_fresh_5k |
| 10x Genomics | 2017 | 10x scRNA-seq on 1k brain nuclei from an E18 mouse | https://support.10xgenomics.com/single-cell-gene-expression/datasets/2.1.0/nuclei_900 | 10x Genomics Resources, nuclei_900 |
| 10x Genomics | 2017 | 10x scRNA-seq on 2k brain nuclei from an adult mouse (>8 weeks) | https://support.10xgenomics.com/single-cell-gene-expression/datasets/2.1.0/nuclei_2k | 10x Genomics Resources, nuclei_2k |
| ENCODE Project Consortium | 2021 | ATAC-seq from MCF-7 (ENCSR422SUG) | https://www.ncbi.nlm.nih.gov/geo/query/acc.cgi?acc=GSE169929 | NCBI Gene Expression Omnibus, GSE169929 |
| ENCODE Project Consortium | 2021 | ATAC-seq from PC-3 (ENCSR499ASS) | https://www.ncbi.nlm.nih.gov/geo/query/acc.cgi?acc=GSE170337 | NCBI Gene Expression Omnibus, GSE170337 |

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
