## [Editor Report]

De Rop et al. introduce a flexible microfluidics-based single-cell genomics technology that expands and improves previously existing custom droplet-based scRNA-seq protocols (inDrops and Drop-seq) in interesting directions: better data quality, simplified workflow, high-cell recovery, and flexibility towards other single-cell applications.

---

## [Decision Letter]

**Decision letter after peer review:**

Thank you for submitting your article “HyDrop: droplet-based single-cell ATAC-seq and single-cell RNA-seq using dissolvable hydrogel beads” for consideration by *eLife*. Your article has been reviewed by 2 peer reviewers, and the evaluation has been overseen by a Reviewing Editor and Naama Barkai as the Senior Editor. The following individual involved in review of your submission has agreed to reveal their identity: Klaas Mulder (Reviewer #3).

The reviewers are excited about your paper, we will be happy to publish it, but please address the suggestions made below.

*Reviewer #1 (Recommendations for the authors):*

The paper is clearly written and it Includes enough methodological details, for example explaining the rationale behind some of the modifications benchmarked and later selected in the HyDrop protocol. Moreover, the authors exemplify the application of HyDrop to different contexts (e.g. low-input samples). For all these reasons, I think this is an outstanding candidate for publication in *eLife*.

*Reviewer #2 (Recommendations for the authors):*

Specific comments for the authors:

1) The introduction (line 38) contains the following statement "Primers carried or released by the bead allow each individual cell’s mRNA to be indexed inside the droplet".

This is indeed the case for the inDrop method, however is incorrect for Drop-seq, where the RT reaction takes place *after* emulsion breaking. Therefore, this sentence does not represent both methods accurately. However, this distinction is important to make as these are the main methods HyDrop is compared to.

2) Figure 1 and Figure 1—figure supplement 2. The produced hydrogel beads do not seem perfectly monodisperse and uniform in their fluorescent signal after barcode production. It might be good to mention this in the text and briefly discuss whether this will potentially impact the detection of heterogeneity in the cell population after sequencing, or not.

3) Figure 1—figure supplement 6c, the 6uM primer beads are concluded to be the best option for the HyDropRNA application. The images of these beads show some punctate intensities. Would the authors like to comment on this briefly? This could raise some questions for the readers, potentially leading to doubts about how to best adopt the method.

4) Figure 4—figure supplement 1b, please indicate in the legend what the colors encode.

5) The linear amplification method to add the barcodes to the beads (and in the scATAC workflow) is non-standard, would the authors be able to provide an estimation of the rate (or proportions) of errors introduced into the barcodes during this process? The authors state that 88% of the detected barcodes are in the ‘'whitelist’' (1 mismatch allowed). What is the distribution of mismatches in the 12% of the barcode that were discarded? Would that be of interest to estimate the error rate of the linear amplification step?

6) On the conclusion stated on line 116 on the optimal bead primer concentration: it seems that adding half the concentration of primer (6 instead of 12 uM) leads to 10x less reads. Which concentration would they in the end advice and in which context? (not mentioned clearly).

7) Line 258: This sentence is confusing. Some points they elaborate on (use og GTP/PEG for molecular crowding) others they just refer to other papers (use of LNA). Maybe cut into different sentences with explanation and referral to supplemental figure? The overall conclusion can still be linked to the main figure.

8) Figure 8c, It seems from the methods section that the filtering of the InDrop and Drop-seq data is not the same? What happens you perform the same filtering as with your HyDrop analysis?

9) Line 284: the header states HyDrop-ATAC, whereas the experiment and text are on HyDrop-RNA.

10) Line 327. Were the discussed neuronal cell populations also detected in the in-house generated Drop-seq experiments? It would be good to include a mention/discussion of this in the text.

---

## [Author Response]

Reviewer #2 (Recommendations for the authors):Specific comments for the authors:1) The introduction (line 38) contains the following statement: "Primers carried or released by the bead allow each individual cell’s mRNA to be indexed inside the droplet".This is indeed the case for the inDrop method, however is incorrect for Drop-seq, where the RT reaction takes place after emulsion breaking. Therefore, this sentence does not represent both methods accurately. However, this distinction is important to make as these are the main methods HyDrop is compared to.

We thank the reviewer for pointing out this mistake. We have clarified and modified the text to the following:

“Barcoding primers carried by the beads inside the emulsion are then used to index each individual cell’s mRNA. This process occurs either inside the droplet, where the cell’s mRNA is reverse transcribed using barcoded primers released by the barcoded bead (inDrop) or after emulsion breaking, where the cellular mRNA is anchored onto barcodes carried by resin beads (Drop-seq).”

2) Figure 1 and Figure 1—figure supplement 2. The produced hydrogel beads do not seem perfectly monodisperse and uniform in their fluorescent signal after barcode production. It might be good to mention this in the text and briefly discuss whether this will potentially impact the detection of heterogeneity in the cell population after sequencing, or not.

While the small images in Figure 1 and Figure 1 supplement 2 indeed give the impression that the beads are heterodisperse in both size and fluorescent signal, data generated by our automated image analysis script indicates that the diameter of our beads vary by < 5% of the mean, and the intensity varies by <8% of the mean. To make this more clear to the reader, we have appended the absolute values of 1 standard deviation to both intensity and bead diameter in the supplementary data tables. For more details, the script, fluorescence images and resulting tables are available in GitHub: https://github.com/aertslab/hydrop_data_analysis/tree/main/manuscript_beadqc.

3) Figure 1—figure supplement 6c, the 6uM primer beads are concluded to be the best option for the HyDropRNA application. The images of these beads show some punctate intensities. Would the authors like to comment on this briefly? This could raise some questions for the readers, potentially leading to doubts about how to best adopt the method.

Indeed, some intense spots are present in some micrographs shown in the manuscript. However, we are almost certain that these small spots are dye precipitates or crystal aggregates arising from the primer suspension. We have also observed this while performing the same QC method on 10x genomics beads. For example, Author response image 1 is a QC image of 10x RNA beads.

**Author response image 1. sa2fig1:** 

In order to clarify this, we have added the following line to the methods:“Occasional punctate intensities (diameter ~1 μm) were observed in quality control performed on both HyDrop and 10X Genomics beads and are thought to be either dye precipitate (crystals aggregates in the oligonucleotide-fluorophore suspension) auto-fluorescent particles.”

4) Figure 4—figure supplement 1b, please indicate in the legend what the colors encode.

Thank you for pointing this out. We have added the following line to the figure caption:

“Color scale encodes each count’s local Gaussian kernel density estimation score.”

This should clarify the legend scores to the reader.

5) The linear amplification method to add the barcodes to the beads (and in the scATAC workflow) is non-standard, would the authors be able to provide an estimation of the rate (or proportions) of errors introduced into the barcodes during this process? The authors state that 88% of the detected barcodes are in the ‘'whitelist’' (1 mismatch allowed). What is the distribution of mismatches in the 12% of the barcode that were discarded? Would that be of interest to estimate the error rate of the linear amplification step?

Thank you for this interesting suggestion. We have calculated the distribution of mismatches in the remaining 12% of unidentified barcode sequences up to a hamming distance of 3. This analysis showed that an additional 1.16% and 1.13% of all barcodes can be assigned to the whitelist when 2 and 3 mismatches are allowed. When hamming distance is increased to 4 and beyond, matching of a barcode to more than 1 whitelist entry becomes possible and obfuscates results. We believe that the remaining ~10% of unidentified barcode sequences are a result of incorporation errors in the oligonucleotide production process. Importantly, in order to limit cost of the barcode plates we opted for standard desalting purification when purchasing our barcode oligonucleotides. Here, the error in base incorporation is about: 1 in 1000 bases (https://eu.idtdna.com/pages/support/faqs/why-does-my-sequence-data-have-errors) which is far more frequent than the reported error rate of the polymerase we use (Kapa HiFi has an error rate of around 1 in 3.6M bases, https://rochesequencingstore.com/wp-content/uploads/2017/10/KAPA-HiFi-PCRKit_KR0368-%E2%80%93-v13.19.pdf) and results in a frame shift. These frameshifted barcodes produce a barcode read that cannot be identified using simple hamming distance based error correction.

We have added the following text to the manuscript:

“An additional 1.16% and 1.13% of HyDrop barcodes could be assigned to the whitelist when 2 and 3 mismatches were allowed respectively. The remaining 9% of barcodes sequences could not be identified and are most likely a result of frame-shift due to incorporation errors in the synthetic oligonucleotide production process. Notably, only 39.7% of barcode reads could be matched with the whitelist in a public inDrop dataset (3), when 1 mismatch was allowed. When pre-filtered cell barcode sequences were used as a whitelist for a public mouse retina Drop-seq dataset (2), 49.5% of reads could be matched.”

6) On the conclusion stated on line 116 on the optimal bead primer concentration: it seems that adding half the concentration of primer (6 instead of 12 uM) leads to 10x less reads. Which concentration would they in the end advice and in which context? (not mentioned clearly).

This was indeed not mentioned strongly enough. We are continuously evaluating the optimal concentration of primers needed for bead synthesis of different HyDrop assays. Based on our experience we have found that the 12 μm HyDrop beads provide the most sensitive performance on both the HyDrop assays. Therefore, we used (and continue to use) 12 µM primer concentration to produce barcoded hydrogel beads for both HyDrop-RNA and HyDrop-ATAC experiments.

We have clarified line 116 to:

“The bead primer concentration with an optimal balance between assay sensitivity and library purity was found to be 12 µM for both scATACseq and scRNA-seq (Figure 1 – Figure supp. 6).”

7) Line 258: This sentence is confusing. Some points they elaborate on (use og GTP/PEG for molecular crowding) others they just refer to other papers (use of LNA). Maybe cut into different sentences with explanation and referral to supplemental figure? The overall conclusion can still be linked to the main figure.

We thank the reviewer for pointing out that this part of the manuscript was not clearly written. We have rewritten the paragraph to:

“To improve the assay’s sensitivity, we interrogated the impact of several alterations to the protocol’s reaction chemistry by testing them on a 50:50 human-mouse (human melanoma, mouse melanoma) mixture. We first investigated the use of pooled exonuclease I treatment after reverse transcription to remove unused barcode primers. We reasoned that these unused barcoded primers could potentially prime transcripts during the subsequent bulk ISPCR, leading to a loss of purity of transcripts associated with a given barcode. As evident from the increase in pure cells detected, we found that exonuclease I treatment indeed improved assay purity (Figure supp. 1). We then tested the implementation of a locked nucleic acid (LNA) in the TSO, as it has been shown to increase assay sensitivity due to increased hybridization stability of the TSO-cDNA dimer (30). We also investigated the addition of GTP/PEG to the in-droplet reverse transcription reaction. Both the addition of PEG as a molecular crowding agent and GTP have been shown to improve assay sensitivity in SMART-seq3 (31). Finally, we wondered whether a second strand synthesis library construction approach could outperform the TSO/ISPCR approach. In order to test this, we compared both alkaline hydrolysis and enzymatic treatment (RNAse H) to remove the RNA strand from the first strand product, and evaluated the performance of both Bst 2.0 DNA polymerase and Klenow (exo-) fragment for second strand synthesis (32,33). We found that the classical TSO and ISPCR protocol supplemented with GTP/PEG performed best, yielding a median of 2,110 UMIs and 1,325 genes per cell with a species-purity of 90.1% (Figure 8a, Figure 8 – Figure supp. 1).”

8) Figure 8c, It seems from the methods section that the filtering of the InDrop and Drop-seq data is not the same? What happens you perform the same filtering as with your HyDrop analysis?

We thank the reviewer for commenting on this filtering issue. In our original analysis, the sensitivities of Drop-seq and inDrop were deduced from the expression matrices provided as supplementary material to the respective publications, and these contained only the cells as filtered by the authors. In the Macosko et al., (Drop-seq) manuscript, these cells were selected based on visual detection of a drop in the UMI barcode rank plot. In the Kalish et al., (inDrop) manuscript, an arbitrary filter of barcodes containing > 500 UMIs was used. As opposed to opting for manual filtering methods, HyDrop employs STARsolo’s automated cell filtering algorithm (identical to 10X Genomics cellranger’s cell filtering algorithm) which is adaptive to the distribution of UMIs in cell barcodes. We believe that this automated filter is the best approach for HyDrop, as it is independent of human judgment, matches our expected cell recovery, and clearly distinguishes background from true cells in the barcode rank plots.

We decided to re-analyze the published InDrop and Drop-seq datasets starting from the raw fastq files, using STARsolo for cell filtering. Surprisingly, both for the inDrop and Drop-seq data, the STARsolo filtering method yielded very different results of cell filtering compared to the reported results. STARsolo identified only 29k cells in the

Drop-seq’s 49k cells dataset, and 21k in inDrop’s 31k cells dataset. This discrepancy likely results from the lack of a clear separation between cells and non-cells in inDrop and Drop-seq’s UMI/barcode-rank, which makes it difficult for STARsolo to filter true cells from background. If we were to reduce the number of cells in the inDrop and Dropseq samples by such large margins, then the fraction of raw reads that end up as unique transcript counts in barcodes identified as cells, would fall too low.

As an alternative comparison of the three data sets, we decided to retain the number of cells reported in the Macosko et al., manuscript and sampled the sequencing data to HyDrop’s depth (52738 reads/cell). For the inDrop data, this constraint led to a reduction of the number of cells from 31293 to 27094 and resulted in a minimum UMI cutoff of 577 instead of the original 500 for the Kalish et al., inDrop data. As a result, the dataset’s median genes/UMIs increased from 1097/1521 to 1321/1920. Note that we also allowed 1 mismatch in Drop-seq’s barcodes (like with inDrop and HyDrop), a step which is not performed in the original Drop-seq analysis. This analysis is transparently reported step-by-step in our GitHub repository at: https://github.com/aertslab/hydrop_data_analysis/tree/main/HyDropRNA_publicdata_comparison.

We have also adjusted the figure and adapted the main text as following:

“We then used HyDrop-RNA to generate 9,508 single nuclei transcriptomes from snap-frozen mouse cortex in a single experiment. At a sequencing saturation of approximately 60% duplicates and with 86% of reads mapping to genome and 55% of reads mapping to transcriptome, we achieve a median of 3,389 UMIs and 1,658 genes per cell before, and 3,404 UMIs and 1,662 genes per cell after doublet filtering (Figure 8c), compared to the median of 1,920 UMIs and 1,321 genes detected by inDrop snRNA-seq on mouse auditory cortex neurons (3) and the median of 1,071 UMIs and 763 genes reported by Drop-seq on mouse retina neurons (2). Both datasets also exhibited a lower read alignment efficiency, with Drop-seq mapping 52%/21%, and inDrop mapping 48%/28% of reads to the mouse genome/transcriptome. Importantly, HyDrop’s “sequencing efficiency” is higher than both inDrop and Drop-seq’s: 7.44% of HyDrop’s sequenced reads end up as mapped transcripts with a unique molecular identifier, whereas this metric is 3.24% for the inDrop dataset and 4.46% for the Drop-seq dataset (Figure 8 —figure supplement 2). 10x Chromium v2 gene expression reference data reports a median number of genes of 775-2,679 and a median number of UMIs of 1,127-6,570 on E18 and adult mouse brain nuclei (see methods).”.

We also added the following to the methods section:

“Raw inDrop (SRR10545068 to SRR10545079) and Drop-seq (SRR1853178 to SRR1853184) sequencing data was downloaded from SRA. Data was sampled to the HyDrop-RNA mouse cortex sample sequencing depth (52738 reads per cell) and mapped to mouse reference genome using STARsolo, allowing 1 mismatch in the cell barcodes (like in our HyDrop-RNA and HyDrop-ATAC analyses). Full analysis process is documented on the Hydrop GitHub repository (https://github.com/aertslab/hydrop_data_analysis/tree/main/HyDropRNA_publicdata_comparison).”

9) Line 284: the header states HyDrop-ATAC, whereas the experiment and text are on HyDrop-RNA.

We thank the reviewer for this remark and have corrected it in the final manuscript.

10) Line 327. Were the discussed neuronal cell populations also detected in the in-house generated Drop-seq experiments? It would be good to include a mention/discussion of this in the text.

Previous analysis of these drop-seq experiments did not yield such a granularity in detected cell types, separating mostly glia and neurons. We believe that this difference can be explained partially due to the lower data quality, but mostly due to the difference in sample: the HyDrop runs were performed on specifically FAC-sorted neuronal populations, whereas the Drop-seq runs were performed on unsorted whole brain. We clarified in the text that the cell populations found by HyDrop after FAC sorting a specific subset are not expected to be found in the Drop-seq whole-brain dataset as follows:

“In comparison, in-house Drop-seq performed on fly brain neurons the entire fly brain recovered a median of 579 UMIs and 289 genes per cell at a deeper sequencing saturation (11).”